# Structural variant-based pangenome construction has low sensitivity to variability of haplotype-resolved bovine assemblies

Alexander S. Leonard [1✉], Danang Crysnanto [1], Zih-Hua Fang [1], Michael P. Heaton [2],
Brian L. Vander Ley [3], Carolina Herrera[4], Heinrich Bollwein[4], Derek M. Bickhart[5], Kristen L. Kuhn[2],
Timothy P. L. Smith [2], Benjamin D. Rosen [6✉] & Hubert Pausch [1✉]

Advantages of pangenomes over linear reference assemblies for genome research have recently been established. However, potential effects of sequence platform and assembly approach, or of combining assemblies created by different approaches, on pangenome construction have not been investigated. Here we generate haplotype-resolved assemblies from the offspring of three bovine trios representing increasing levels of heterozygosity that each demonstrate a substantial improvement in contiguity, completeness, and accuracy over the current *Bos taurus* reference genome. Diploid coverage as low as 20x for HiFi or 60x for ONT is sufficient to produce two haplotype-resolved assemblies meeting standards set by the Vertebrate Genomes Project. Structural variant-based pangenomes created from the haplotype-resolved assemblies demonstrate significant consensus regardless of sequence platform, assembler algorithm, or coverage. Inspecting pangenome topologies identifies 90 thousand structural variants including 931 overlapping with coding sequences; this approach reveals variants affecting *QRICH2*, *PRDM9*, *HSPA1A*, *TAS2R46*, and *GC* that have potential to affect phenotype.

[1] Animal Genomics, ETH Zurich, Universitaetstrasse 2, 8006 Zurich, Switzerland. [2] U.S. Meat Animal Research Center, USDA-ARS, 844 Road 313, Clay Center, NE 68933, USA. [3] Great Plains Veterinary Educational Center, University of Nebraska-Lincoln, Lincoln, NE 68588, USA. [4] Clinic of Reproductive Medicine, Department for Farm Animals, University of Zurich, 8057 Zurich, Switzerland. [5] Dairy Forage Research Center, USDA-ARS, 1925 Linden Drive, Madison, WI 53706, USA. [6] Animal Genomics and Improvement Laboratory, USDA-ARS, 10300 Baltimore Ave, Beltsville, MD 20705, USA.
✉email: alexander.leonard@usys.ethz.ch; ben.rosen@usda.gov; hubert.pausch@usys.ethz.ch

Cattle are a substantial component of global animal-based food production, and are raised for meat, milk, or both. Two subspecies of cattle, taurine and indicine, have emerged from at least two distinct domestication events[1,2], with artificial selection for production goals or environmental adaptation contributing to diversity within cattle, resulting in the current existence of hundreds of taurine and indicine cattle breeds. Interbreeding and introgressions with other bovids, like yak and banteng[3,4], further drive an increase in genetic diversity within Bovinae.

The *Bos taurus* reference genome was first drafted in 2004 and was based on whole-genome shotgun sequence of a Hereford cow, supplemented with sequences of bacterial artificial chromosome clones prepared from DNA of her sire[5]. A major revision using Pacific Biosciences continuous long read (CLR) sequencing of the same cow was recently reported (ARS-UCD1.2 [6]) and remains the accepted reference for conducting genomic studies in cattle due to extensive annotation efforts and connections to historical analyses, despite more recent bovine assemblies having higher contiguity and accuracy[7–9].

Sequence variability between cattle breeds at both the single nucleotide (SNP) and short insertion or deletion (indel) level has been extensively characterized through reference-guided approaches[10–12]. Short sequencing reads have also been used to study larger structural variants (SVs) and variation located in repetitive or challenging regions across domestic cattle[13–15], although their accuracy is limited compared to using long sequencing reads[16]. SVs may be involved in expression quantitative trait loci more often than previously estimated, and can impact gene expression more than shorter variants[17]. However, previous studies suffer from potential reference bias, because the use of a single taurine cattle reference assembly fails to reflect the immense genomic diversity present in global breeds of domestic cattle[18–21].

Pangenomes have long been proposed[22] as a way to better reflect variation present in a group of individuals (e.g., breed, species, clade, etc.). Pangenomes can be constructed from variants called through reference-guided approaches[23,24], contigs assembled from reads which failed to align to the reference[25], or multiple whole-genome assemblies[26,27]. The latter approach may more faithfully capture challenging regions and SVs due to the complexity of calling and representing nested variation. Third-generation sequencing continues to become more cost-effective and accessible, making population-scale de novo assemblies more feasible. An influx of high-quality assemblies makes the need for pangenome representations more pressing, although the effects of integrating disparate assemblies into pangenomes are unknown.

In this work, we combine de novo assembled genomes from three bovine trios of varying heterozygosity into structural variant-based pangenomes. Haplotype-resolved, reference-quality assemblies are generated from the trio offspring using Pacific Biosciences high-fidelity (HiFi [28,29]) and Oxford Nanopore Technologies (ONT [30]) sequencing and various assembly algorithms. We use this set of assemblies from disparate sequencing and origin to evaluate the effects of assembly approach, or a combination thereof, on pangenome construction. We then demonstrate the utility of a bovine pangenome through analyses using SVs to assess evolutionary relationships between Bovinae, recovering putatively trait-associated SVs, and quantifying SV-coding sequence overlaps.

## Results

The three examined bovine trios (Fig. 1a–c) reflect diverse breeding strategies (within-breed, inter-subspecies, and inter-species) and increasing heterozygosity. The first F1 (OxO) was a cross between two Original Braunvieh cattle (*Bos taurus taurus*), but was still substantially less inbred compared to the cow used for the ARS-UCD1.2 reference (pedigree-based coefficient of inbreeding of 0.07 compared to 0.30[6], respectively). The second F1 (NxB) was a cross between Nellore (*Bos taurus indicus*) and Brown Swiss (*Bos taurus taurus*) cattle and the third F1 (GxP) was an interspecies cross between a gaur (*Bos gaurus*) bull and a Piedmontese (*Bos taurus taurus*) cow. HiFi and ONT reads were collected for each F1, and short reads were collected for all animals in the trios (Fig. 1d, Supplementary Table 1). F1 long reads were separated into paternal, maternal, and unknown origin[31]. Separability, the proportion of reads assigned to a parent-of-origin bin, improved significantly from 81.1% to 99.9% with increasing heterozygosity for HiFi reads, but was near perfectly separable for ONT at all examined heterozygosities (Fig. 1e, Supplementary Table 2).

**Assembling bovine genomes.** Investigating the resiliency of pangenome graphs to variable input assemblies was initiated by examining the properties of each type of assembly and their sensitivity to input data parameters. Contigs were assembled for each F1 using multiple assemblers for HiFi and ONT data (Fig. 1f, methods). Contigs were scaffolded by alignment to ARS-UCD1.2 to produce the final assemblies, an approach with minimal bias given the well-curated reference and highly contiguous assemblies[32,33]. The gaur has a centromeric fusion of chromosomes 2 and 28 resulting in a different karyotype than that of domestic cattle[34], but this scaffolding approach was still successful for the paternal haplotype of the GxP (Supplementary Fig. 1a). The haplotype-resolved assemblies for the inter-subspecies NxB and interspecies GxP are directly breed/species-specific, while both haplotypes of the OxO constitute the same breed, resulting in six haplotype-resolved assemblies, five of which represent novel breeds/species. For breed/species-level analysis, we only use the maternal haplotype of the OxO by choice, while analyses displaying intra-breed variation use both haplotypes (methods). The "quality" of each assembly was assessed by several widely used metrics: contiguity (NG50) representing the size distribution of contigs, phasing (PG50) to characterize haplotype separation, correctness (QV) quantified as Phred-scaled base-error rate, and completeness (BUSCO) which approximates the percentage of near-universal, single-copy genes that were identified. These metrics provide a coarse-grained summary of assembly quality amenable to comparisons.

The assemblies produced by hifiasm for HiFi data and Shasta for ONT data were selected for further analysis based on having the best quality metrics (Table 1) and computational tractability compared to four other tested assemblers (Supplementary Table 3). These assemblies were of reference quality with every examined metric exceeding those of the current *Bos taurus* Hereford-based reference. Improvements over the current reference are substantial, including a 3-11× reduction in autosomal gaps, 1.8-3.6× increase in NG50, and 3-22× reduction in base errors. Furthermore, they all exceed the current standards set by the Vertebrate Genomes Project (VGP)[35]. The ONT-based assemblies were marginally above the QV targeted by the VGP, but other metrics for these assemblies such as the contig or phased contiguity are orders of magnitude greater than VGP thresholds.

The HiFi- and ONT-based assemblies were generally comparable, however there were notable differences in the average assembly correctness and genome size metrics. Shasta assemblies averaged QV 41.5 after one round each of polishing with ONT reads and short reads, while hifiasm assemblies reach QV 47.6 without any polishing. The log scale of QV means that the hifiasm

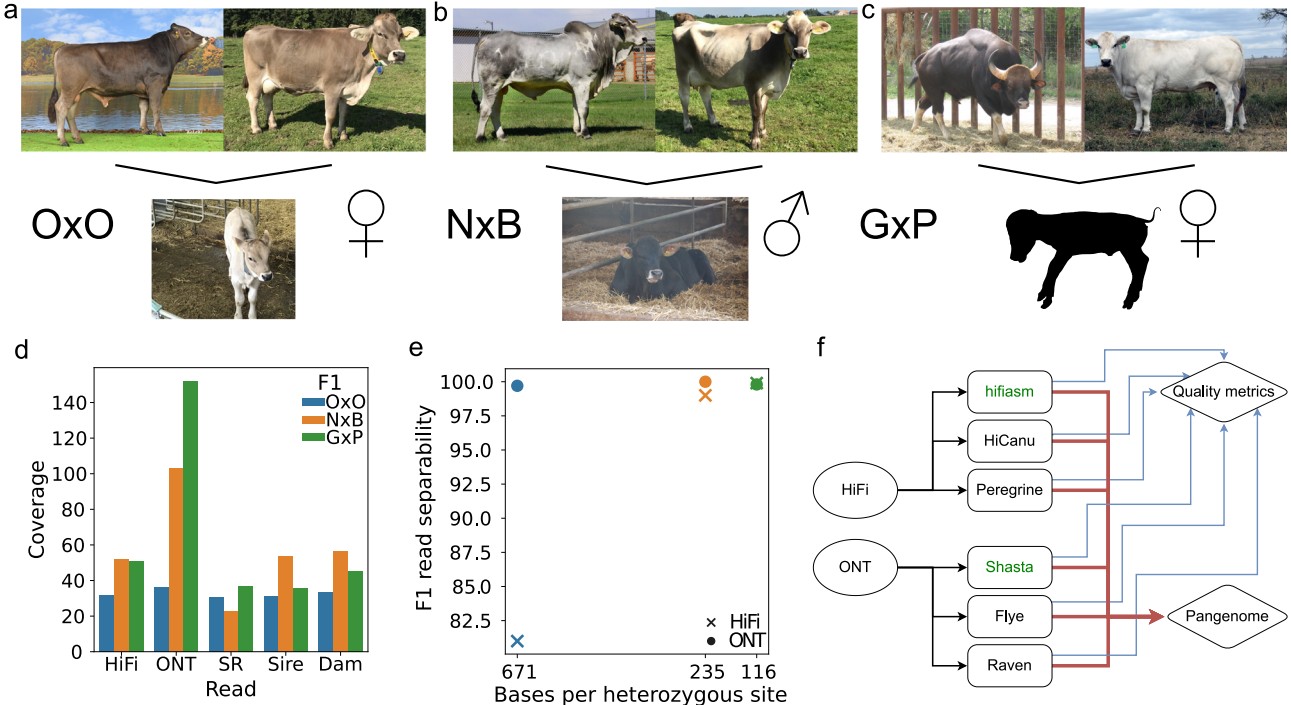

**Fig. 1 Overview of bovine trios. a–c** Representative animals for the parents of the three bovine trios and the respective F1s (OxO, NxB, and GxP) examined in this study. The OxO and GxP were female, while the NxB was male. **d** The three respective F1s were sequenced to 32-, 52-, and 51- fold HiFi coverage, with read N50 of 20, 21, and 14 Kb. ONT sequencing was performed to 36-, 103-, and 152-fold coverage respectively, with read N50 of 65, 45, and 49 Kb. Coverage is determined with respect to an assumed genome size of 2.7 Gb. F1 short reads (SR) were collected to 31-, 23-, and 37-fold coverage. **e** Separating F1 reads into parental haplotype bins improved with increasing heterozygosity for HiFi, but F1 reads were nearly 100% separable for ONT even at low heterozygosity (color from panel **d**). **f** HiFi reads were assembled with hifiasm, HiCanu, and Peregrine, while ONT reads were assembled with Shasta, Flye, and Raven. Green font indicates the tools used to produce the assemblies that are discussed in detail. Assemblies were assessed individually for quality metrics (blue lines) as well as integrated together into pangenome analyses (red lines).

**Table 1 Quality metrics for ten haplotype-resolved assemblies.**

| Breed or species | Haplotype | Read technology | Size (autosomal size) | Contigs (autosomal contigs) | NG50 | PG50 | QV | BUSCO (single-copy) | Repeat |
|---|---|---|---|---|---|---|---|---|---|
| Original Braunvieh | Paternal (X) | HiFi | 3.15 (2.57) | 2108 (107) | 56.0 | 16.2 | 49.2 | 95.7 (93.9) | 49.39 |
| | Paternal (X) | ONT | 2.70 (2.48) | 2622 (109) | 71.6 | 2.8 | 40.7 | 95.2 (93.5) | 43.27 |
| | Maternal | HiFi | 3.11 (2.57) | 1706 (105) | 47.0 | 23.6 | 49.7 | 95.7 (93.9) | 48.95 |
| | Maternal | ONT | 2.70 (2.48) | 2622 (109) | 71.6 | 2.7 | 40.3 | 95.1 (93.4) | 43.19 |
| Nellore | Paternal (Y) | HiFi | 2.95 (2.60) | 1217 (52) | 94.4 | 79.1 | 46.1 | 93.3 (91.8) | 47.81 |
| | Paternal (Y) | ONT | 2.57 (2.49) | 1457 (67) | 68.5 | 64.9 | 42.4 | 92.8 (91.3) | 42.64 |
| Brown Swiss | Maternal | HiFi | 3.07 (2.62) | 1045 (58) | 86.7 | 81.1 | 45.6 | 95.9 (94.2) | 48.43 |
| | Maternal | ONT | 2.67 (2.48) | 1268 (71) | 64.0 | 53.0 | 42.5 | 95.3 (93.7) | 42.85 |
| gaur | Paternal (X) | HiFi | 3.02 (2.52) | 1352 (75) | 73.5 | 61.2 | 48.4 | 95.7 (94.1) | 47.73 |
| | Paternal (X) | ONT | 2.64 (2.48) | 532 (89) | 68.1 | 68.1 | 41.2 | 95.1 (93.3) | 42.26 |
| Piedmontese | Maternal | HiFi | 3.10 (2.56) | 1427 (90) | 52.0 | 47.6 | 48.3 | 95.8 (94.1) | 48.43 |
| | Maternal | ONT | 2.66 (2.48) | 782 (64) | 82.8 | 82.8 | 40.9 | 95.3 (93.6) | 43.06 |
| Hereford (ARS-UCD1.2) | (N/A) | CLR | 2.72 (2.49) | 2597 (289) | 25.9 | N/A | 35.8 | 95.7 (93.9) | 42.96 |
| VGP Standards | | | | | 1 | 0.1 | 40 | 90 | N/A |

The assembly haplotype is either maternal or paternal (indicating either an "X" or "Y" paternal sex chromosome). The ARS-UCD1.2 reference is not haplotype-resolved and lacks sufficient parental data to assess phasing, hence the N/A. Size and contigs refer to the entire genome assembly, while the autosomal values only measure chromosomes 1 through 29. NG50 is the contig N50 using the ARS-UCD1.2 reference sequence as the expected length. PG50 is NG50 after splitting contigs into haplotype-phased blocks. Phasing and QV are determined through merqury using parental and F1 short reads. Scaffolded NG50 is not shown, as all assemblies are effectively end-to-end (excluding centromeres and telomeres), with values greater than 100 Mb. Assemblies are available online[107].

assemblies had a 4-fold reduction in base errors compared to the Shasta assemblies, indicating the ability of HiFi data to achieve higher quality in fewer steps. In contrast, phasing in Shasta assemblies is better compared to hifiasm, while both platforms showed improved phasing at higher heterozygosity in agreement with the relative ability to sort F1 reads by parental origin prior to

assembly. There was insufficient ONT coverage for the OxO to perform the bin-then-assemble approach used for the NxB and GxP and so instead a diploid polishing approach was used to phase the haplotypes (methods), resulting in reduced PG50.

The mean autosomal genome size of the assemblies generated by hifiasm and Shasta was $2.57 \pm 0.03$ Gb and $2.48 \pm 0.004$ Gb

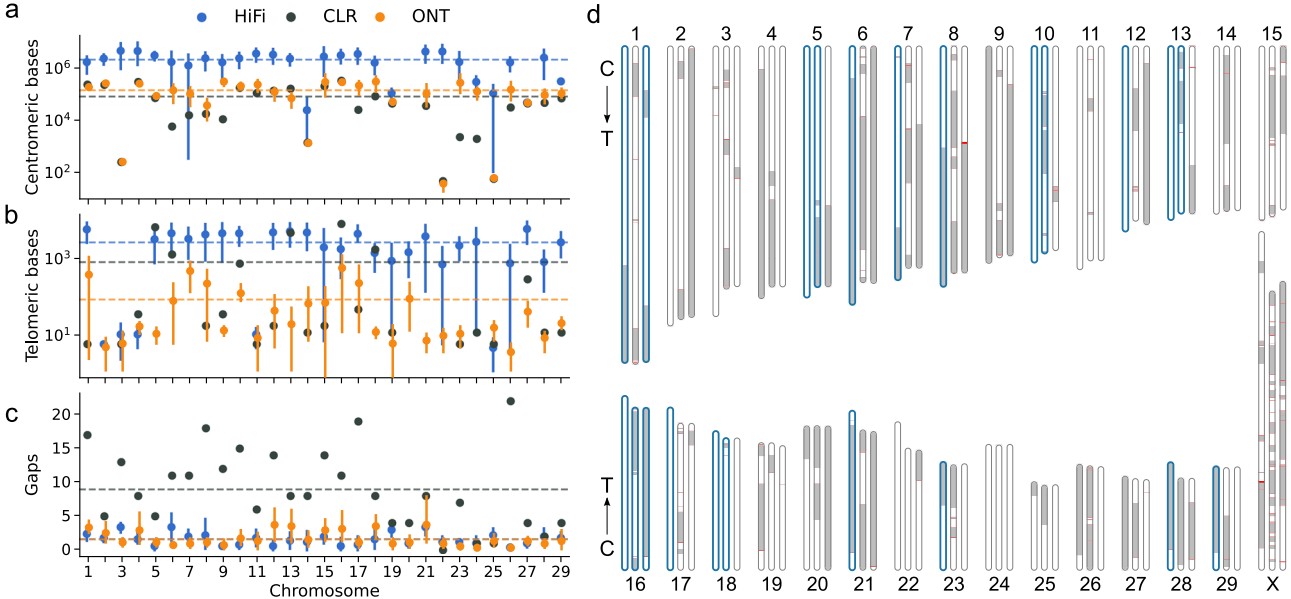

**Fig. 2 Centromeric and telomeric completeness of assemblies produced by hifiasm and Shasta. a** The mean number of bases identified per autosome as "Satellite" by RepeatMasker for the $n = 5$ hifiasm (blue) and $n = 5$ Shasta (orange) assemblies, where error bars indicate the 95% confidence interval. The black dots represent values from the CLR-based ARS-UCD1.2. Dashed lines indicate the autosome-wide mean for the respective color of points. Mean values of 0 (e.g., chromosome 20) are not shown due to the log scale. **b** Similar to (**a**), but the number of bases in telomeric repeats within 10 Kb of chromosome ends. **c** Similar to (**a**), but the number of scaffold gaps. **d** Chromosome ideograms for ARS-UCD1.2 (center), and Brown Swiss assemblies produced by hifiasm (left) and Shasta (right). Scaffolded contigs alternate white/gray across gapped regions, which are colored red. Chromosomes which are predicted to extend from centromere to telomere are bolded in blue, of which 7 and 3 are also gapless for hifiasm and Shasta respectively. Arrows indicate the centromere (C) to telomere (T) directionality of the chromosomes (this applies only to autosomes, as the X chromosome is submetacentric).

respectively, such that on average each hifiasm autosome was 2.8 Mb longer than ARS-UCD1.2, and 0.34 Mb shorter for Shasta. The additional length of hifiasm autosomes was primarily due to the presence of more repetitive sequences (43.5% repetitive content in hifiasm versus 41.7% in Shasta), especially centromeric repeat sequence which accounted for 56% of the additional sequence compared to Shasta assemblies. Previous studies have shown that the higher accuracy of HiFi reads allows assemblers to confidently assemble through more repeats despite having shorter read lengths[36], leading to extension of autosomal contigs into flanking centromere sequence. Hifiasm assemblies also contained more sequence in contigs not assigned to chromosomes (average of 300 Mb) compared to Shasta assemblies (50 Mb). These unassigned contigs were composed primarily of repetitive sequences (Supplementary Table 4) including novel centromeric sequence and long terminal repeats. Unplaced contigs were generally higher in repeat content than the scaffolds (88% versus 48% for hifiasm and 87% versus 42% for Shasta), and thus would present a challenge to scaffolding by any technology including the reference alignment approach applied here.

Bovine autosomes are acrocentric[37], and so a complete bovine assembly is conceptually closer to "centromere-to-telomere". Hifiasm assemblies contained substantially more centromeric sequence than Shasta assemblies, respectively averaging 2.01 and 0.14 Mb per autosome, compared to 0.08 in the ARS-UCD1.2 reference (Fig. 2a). Similarly, hifiasm autosomes average 2.6 Kb of vertebrate telomeric repeats (TTAGGG) within 10 Kb of the chromosome end, compared to 0.8 Kb for the ARS-UCD1.2 reference. Telomeres were almost entirely missing in Shasta assemblies, averaging only 88 bp of telomeric repeats per autosome (Fig. 2b), a reported issue with ONT basecalling[38]. Chromosomes which contain at least 50 kb of centromeric repeats at the proximal end and 500 bp of telomeric repeats at the distal end are considered to be end-to-end (but not necessarily "complete"), of which there were 5 for ARS-UCD1.2, and a

mean of 13.2 and 1.2 for hifiasm and Shasta across the five breeds/species. Hifiasm and Shasta assemblies had a near equal distribution of gaps, averaging about 1.5 gaps per autosome, compared to nearly 9 in the ARS-UCD1.2 reference (Fig. 2c). These observations hold in general for all HiFi- and ONT-based bovine assemblies investigated (Supplementary Fig. 2). These differences are visible on the example Brown Swiss chromosome ideograms in Fig. 2d. Both HiFi- and ONT-based assemblies were able to assemble the 16 Kb bovine mitochondrial DNA[39].

**Optimal sequencing coverage depths**. Impact of sequencing depth on assembly quality was assessed using the Brown Swiss haplotype of the NxB as an example. Subsets of the 52× diploid HiFi and 55× haploid (trio binned) ONT reads, respectively, were randomly sampled to mimic lower sequencing depths (Fig. 3). Completeness metrics (e.g., BUSCO or k-mer content) plateau when coverage increased above 25-fold, however other metrics like contiguity or correctness continued to benefit from higher sequencing coverage with diminishing returns. The trio aware mode of hifiasm only required about 19× diploid coverage of HiFi reads to meet the VGP targets. Shasta required around 28× haploid coverage (corresponding to roughly 56× diploid coverage) to achieve the necessary QV, although 17× haploid (34× diploid) coverage fulfills the contiguity and completeness targets. The minimum VGP-satisfying coverage varies slightly for the different F1s due to different input sequencing read properties but is approximately consistent across all examined bovine trios (Supplementary Fig. 3).

The higher accuracy of HiFi reads allows hifiasm to exploit sequence common to both haplotypes during trio aware assembly. Reaching a comparable quality through a trio binned approach required approximately 16% higher coverage, with 11× haploid coverage (22× diploid) necessary. The higher error rate of ONT reads makes haplotype-aware correction and phasing

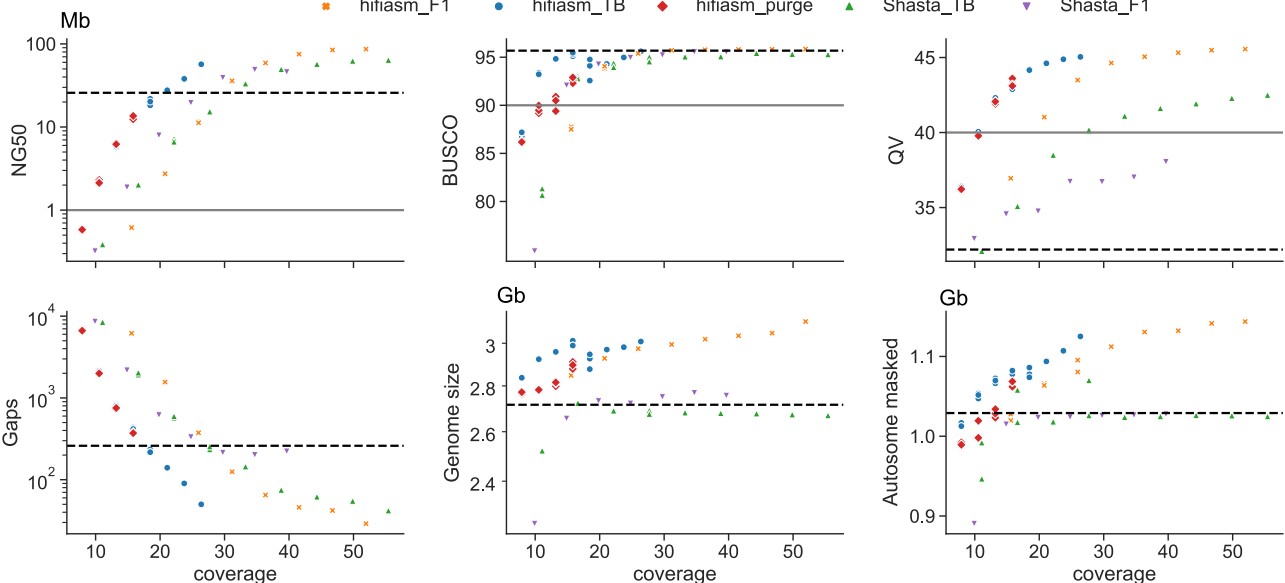

**Fig. 3 Assembly quality at subsampled coverages.** Trio aware hifiasm (hifiasm_F1) uses diploid coverage while Shasta (Shasta_TB) uses haploid coverage. We additionally examined trio binned hifiasm (hifiasm_TB) using haploid coverage and the polish-phased Shasta approach using diploid coverage (Shasta_F1). NG50, BUSCO, QV, autosomal gaps, and genome size are defined in Table 1, while autosomal masked is the number of autosomal bases within repetitive elements as identified by RepeatMasker. The black dashed line represents the relevant value for the ARS-UCD1.2 reference and the gray solid line is the VGP target where applicable. Three subsampling replicates were performed for lower coverage assemblies (<20x for HiFi and <30x for ONT) due to their higher stochasticity. For trio binned hifiasm assemblies below 15x coverage, we manually set the duplication purging parameter (hifiasm_purge) and reran on the same subsamplings.

challenging, and so diploid assembly followed by haplotype separation or diploid-aware polishing is less effective than trio binning (Fig. 3, up triangles versus down triangles). Phasing is particularly poor, with the PG50 25 times smaller on average compared to trio binning approaches.

There is an increased risk of coverage gaps when sequencing coverage is reduced, even if the resulting assemblies achieve certain genome-wide standards. When 11× HiFi coverage is aligned to ARS-UCD1.2 and binned into 10 Kb windows, there are 950 regions on average of near total dropout (<1× coverage) across the autosomes. This drops by 23% at 13.5× coverage and by 30% at 16× coverage, as the effects of stochasticity are reduced. While the overall assembly quality does not fluctuate substantially between random subsamplings (Fig. 3), it may overestimate the quality at specific regions. Furthermore, default parameters of assemblers are typically tuned to higher coverage and assembling at low coverage can introduce subtle issues. Hifiasm could underestimate a parameter related to duplication purging at coverages below 15×, resulting in a sharp transition to larger assemblies with more duplicated BUSCO genes (Fig. 3, blue circles versus red diamonds). Manually setting the parameter to its expected value recovered similar behavior seen in higher coverage assemblies.

**Constructing a bovine pangenome.** Sequencing technologies and assemblers evolve rapidly, and so even recently generated bovine assemblies, including the ones reported here, have been produced under non-uniform conditions (e.g.[18,19],). Given the differences we observed between HiFi- and ONT-based assemblies, especially in comparison to the CLR-based ARS-UCD1.2 reference, it was crucial to examine how pangenome construction responded to different assembly inputs.

Pangenomes were constructed with minigraph[26] using ARS-UCD1.2 as the initial backbone of the graph structure. Assemblies were iteratively added into the graph, and regions of synteny were ignored while sufficiently diverged subsequences (>50 bp) instead augmented the graph with new nodes ("bubbles"). Pangenomes for each autosome were constructed from all hifiasm assemblies, all Shasta assemblies, or random mixtures of the two. Graph properties of each pangenome in terms of the amount of non-reference sequence added, were generally robust to the input assemblies (Fig. 4a, b, Supplementary Table 5). Pangenomes constructed from five hifiasm assemblies had more non-reference sequence added compared to five Shasta assemblies (82.5 Mb across 88.5k nodes versus 63.5 Mb across 90.2k nodes), in agreement with the greater completeness observed in hifiasm assemblies. Approximately 92.1% of identified SVs were common between hifiasm and Shasta pangenomes, with 3.6% and 4.3% unique to each respectively (Fig. 4c). There was not a clear bias between HiFi- and ONT-based pangenomes, with only 1.8 and 0.39 Mb non-reference sequence uniquely identified in each. These bubbles were more repetitive than autosome-wide averages (53% and 45% respectively, Supplementary Table 6). Minigraph is sensitive to the order of integration, and on rare occasions constructed significantly different bubbles, particularly for palindromic sequences, resulting in larger variance on chromosomes 7 and 12 (Supplementary Fig. 4).

Pangenomes constructed from lower coverage assemblies remained robust. We selected hifiasm and Shasta assemblies generated with an average of 21.6× diploid and 24.9× haploid coverage respectively, which approximately satisfy the VGP standards. The hifiasm assemblies have a higher average QV compared to the Shasta assemblies (41.9 versus 39.3), but lower NG50 (2.3 Mb versus 19.5 Mb) and more autosomal gaps (1869 versus 323) (Supplementary Table 7). Although these assemblies are substantially worse than their full-coverage counterparts, the resulting pangenomes are similar (Fig. 4a–c). Taking the high coverage SVs as the truth set, the low coverage hifiasm and Shasta pangenomes have an F1 score of 98.5 and 94.7 for SV discovery respectively. The low coverage Shasta pangenomes tend to identify a greater number of SVs not present in other pangenomes, and so may be false positives.

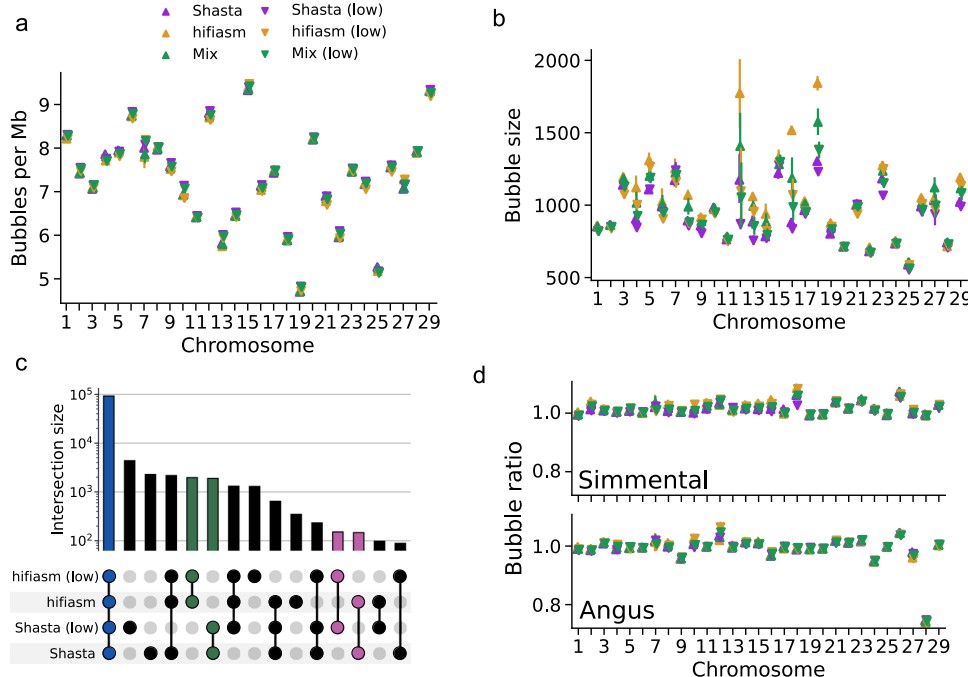

**Fig. 4 Pangenomes are generally robust to different input assemblies. a** The number of large (>1 Kb) bubbles is highly consistent across hifiasm, Shasta, and mixed pangenomes at both full (up triangles) and lower (down triangles) coverages. **b** The mean bubble size is also consistent across different inputs, but bubbles are larger on average in hifiasm pangenomes compared to Shasta pangenomes. **c** The vast majority (84.5%) of SVs identified through minigraph are present in all pangenomes (blue). SVs unique to either hifiasm or Shasta (green) only account for about 3.5% of all SVs, while SVs only identified through either full or lower coverage pangenomes are negligible (pink). **d** Comparing the number of bubbles present in Simmental- or Angus-backed pangenomes to the ARS-UCD1.2-backed pangenome in a) shows consistency. Angus chromosome 28 is the only exception due to its incomplete sequence. All points reflect the mean over 20 stochastic pangenome constructions, and error bars indicate the 95% confidence interval.

Pangenomes constructed using the existing Angus (CLR-based)[40] or Simmental (ONT-based)[8] assemblies as backbones (Fig. 4d) produced similar results compared to using the Hereford-based ARS-UCD1.2. More non-reference sequence was identified in the lower-quality Angus-backed pangenome (+13%), while the more complete Simmental-backed pangenomes had less (−6%). Reference-bias propagates through minigraph's pangenomes, such as the missing sequence in Angus chromosome 28[41] resulting in 25% fewer bubbles compared to using ARS-UCD1.2 (Fig. 4d).

**Quantifying bovine structural diversity through the pangenome**. Potential effects of particular assembly approaches or sequencing technologies on the utility of pangenome graphs for structural variation detection were investigated through the analysis of mutual and private bubbles. Structural variation bubbles were associated with their source assembly by retracing each "haplotype walk" through the graph. The phylogenetic topology of their evolutionary relationship was then estimated by counting the number of mutually exclusive bubbles any two assemblies have to construct a condensed distance matrix. The results are consistent with expectations, with the gaur (*Bos gaurus*) largely separate, followed by the Nellore (*Bos taurus indicus*), and then the three taurine cattle (Fig. 5a). All constructed pangenomes unequivocally predicted the first two branches, while there was also good agreement within the closely related taurine cattle (Fig. 5b).

Several chromosomes (e.g., 2, 6, 9, etc.) are observed to only predict a single taurine topology, even across different assembly inputs and coverages, while other chromosomes (e.g., 1, 12, 16, etc.) have multiple predicted topologies with similar frequencies. This may indicate that certain chromosomes harbor greater

structural variation between specific taurine breeds, which may be reflected in phenotypic differences. We compared against a conventional approach, calling small variants from the parental short reads corresponding to the five haplotypes against ARS-UCD1.2 (Fig. 5b). While the overall topology and magnitude is similar to that found through the pangenome, there was no significant concordance within the taurine cattle, as might be expected given the low linkage disequilibrium between small and structural variants[42].

Genome level dendrograms were mostly consistent across different combinations of inputs, demonstrating that pangenome robustness extends beyond graph properties and into applications. The same general topology was recovered for all examined assemblers, both individually and combined, as well as the lower coverage assemblies. The dendrogram correctly places all assemblies of each breed on their own branch (Fig. 5c) when 31 pangenome assemblies are included (1 reference backbone plus 5 breeds/species × 6 assemblers). Some HiFi- and all ONT-based assemblies for Original Braunvieh are only partially phased due to limited coverage, resulting in greater structural variation within these assemblies (Fig. 5d). The variation between the parental haplotypes of the OxO results in the higher branching point for "O" in Fig. 5c, and highlights the importance of cleanly resolved haplotypes for downstream analyses.

**Pangenome topology at trait-associated SVs**. There presently is no "truth set" of bovine structural variants appearing in the haplotypes represented to benchmark the success of each pangenome graph. Therefore, we selected several previously identified SVs known to impact phenotype which were present in our pangenomes for further investigation. One case is a multi-allelic copy number variation (CNV) at 86.96 Mb on BTA6

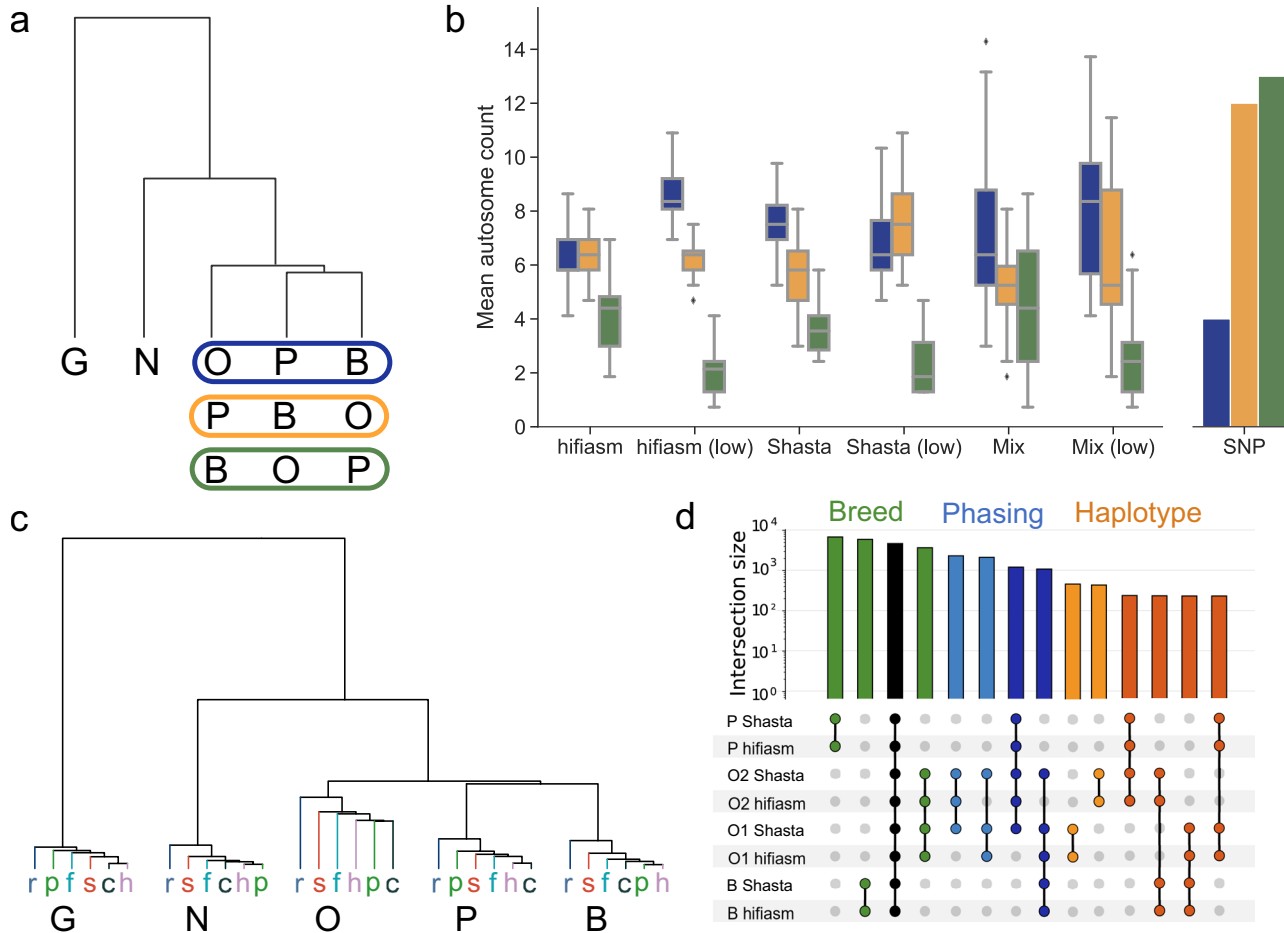

**Fig. 5 SV-based dendrograms. a** All dendrograms followed the same overall topology, with gaur (G) and Nellore (N) clearly differentiated while the taurine cattle displayed three possible arrangements, with either the Original Braunvieh (O), Piedmontese (P), or Brown Swiss (B) more distantly related. The colored boxes represent the three possible orderings of the taurine cattle. **b** Box plots of 20 randomly constructed pangenomes with either all hifiasm, all Shasta, or mixing hifiasm and Shasta assemblies, as well as the low coverage equivalents show good agreement on autosomes displaying a specific topology (color from panel **a**). The box plots represent the median (center line), first and third quartile (box bottom and top), and 1.5x the interquartile range (whiskers). Outliers beyond this range are marked by diamond markers. The SNP dendrogram, based on parental short reads, generally predicted different topologies. **c** Pangenomes including all 30 assemblies from hifiasm (h), Shasta (s), Peregrine (p), Flye (f), HiCanu (c), and Raven (r) predict the same overall topology without ONT or HiFi specific branches. **d** An UpSet plot of a taurine cattle pangenome (Piedmontese, Brown Swiss, and the paternal [O1] and maternal [O2] haplotypes of Original Braunvieh) reveals inter-breed variation (green) as well as intra-breed variation in the Original Braunvieh haplotypes (red & orange). We can also identify phasing error candidates in the Original Braunvieh Shasta assemblies, where SVs are common to both Shasta assemblies but not both hifiasm assemblies (light and dark blue).

encompassing an enhancer of the group-specific component (*GC*) gene. This CNV has pleiotropic effects on mastitis resistance and other dairy traits, and segregates in different breeds of cattle[43,44], although the prevalence of the CNV has not yet been determined for the breeds/species included in our bovine pangenome. All examined assemblers predicted duplication of a 12 Kb segment in Brown Swiss and Original Braunvieh (Fig. 6a, b, Supplementary Fig. 5), although hifiasm required reassembling a small subset of reads as the duplication was missing in the original assembly (methods). The de novo assembled duplicated segments contained various repetitive elements (Fig. 6c) matching those previously reported[43]. There was poor consensus in copy number among the assemblies (Supplementary Fig. 5). Inspection of coverage data retrieved from HiFi- and ONT-binned long read alignments against the ARS-UCD1.2 reference confirmed that the gaur, Piedmontese, and Nellore haplotypes have no additional copies, while the Brown Swiss and Original Braunvieh haplotypes harbor between 2 and 4 additional copies of the 12 Kb segment (Fig. 6d, Supplementary Notes). Coverages derived from the NxB

and OxO F1 short-read alignments are consistent with these values but are unable to resolve the haplotype-specific copy number disagreement observed between HiFi and ONT coverage, as well as between alignments and assemblies (Fig. 6e).

An SV at the *ASIP* gene, encoding the Agouti-signaling protein involved in mammalian pigmentation and located on chromosome 13, was also investigated. This genomic sequence harbors alleles associated with coat color variation in many species including cattle[45,46]. Both Nellore and Brown Swiss cattle present variability in coat color ranging from near white to almost black. The NxB was born with a light coat, which darkened as the bull aged (Supplementary Notes). Our pangenome confirmed great allelic diversity between the bovine assemblies involving insertions and deletions of repetitive elements upstream of *ASIP*. However, the previously described variants associated with coat color variation[45] were not in the pangenomes. Short-read alignments of the Nellore sire confirmed it carried the previously described SV in the heterozygous state, but the F1 inherited the other haplotype (Supplementary Notes). Thus, the darkening of

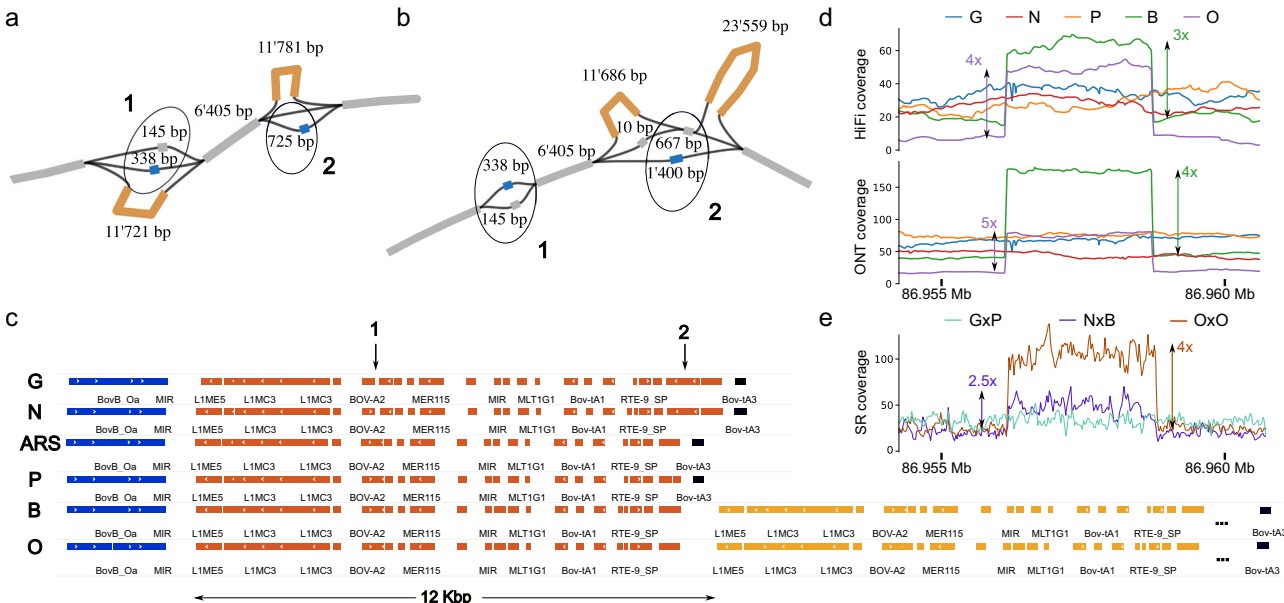

**Fig. 6 Topology of a tandem duplication on BTA6. a, b** Example subgraphs of the promoter region of GC from (**a**) hifiasm- and (**b**) Shasta-based pangenomes respectively. Reference paths (including those in bubbles) are colored gray, while the tandem duplications are orange. Two insertions observed uniquely in Nellore and gaur haplotypes are blue, shown in circles 1 and 2. Complex bubbles generally have suboptimal topologies due to the lack of base-level alignment. For example, the 725 bp insertion is obvious in a), but appears as the difference between a 1400 bp and 667 bp path in (**b**). However, both subgraphs identify the approximately 200 bp (1) and 700 bp (2) insertions in Nellore and gaur, as well as the tandem duplication in Brown Swiss and Original Braunvieh. **c** The 12 Kb repeat structure (orange) is clearly identified by RepeatMasker across all assemblies, shown here for the ARS-UCD1.2 reference and Shasta assemblies for gaur, Nellore, Piedmontese, Brown Swiss, and Original Braunvieh. The two marked gaur/Nellore insertions (1&2) are consistent with the pangenomes in (**a, b**). One additional copy in Brown Swiss and Original Braunvieh is shown (yellow), while the tandem duplication eventually ends with a similar repeat (Bov-tA3, black) to the other assemblies. **d** The identified CNV region shows clear coverage increase in only Brown Swiss and Original Braunvieh, across both HiFi and ONT haplotype-resolved reads, although the HiFi reads suggest one less additional copy than ONT reads. **e** F1 short reads also show increased coverage for the NxB and OxO trios. The NxB coverage increase is consistent with only the Brown Swiss haplotype carrying additional copies.

the coat we observed in the NxB is not due to previously described *ASIP* alleles.

We identified 808 and 845 genes in the ARS-UCD1.2 genome annotation (Refseq release 106) whose coding regions overlap bubbles with complete haplotype information in hifiasm and Shasta-based pangenomes respectively, with 722 common to both (Supplementary Fig. 6, Supplementary Data 1). This overlap is less than expected under random distribution of the same bubbles (one-sided Wilcoxon test: $p = 6.9e{-}10$, methods), as expected given the greater evolutionary conservation of genic regions. Out of 229 affected genes which had pLI scores in human orthologues[47], 35 and 36 overlapping bubbles in hifiasm and Shasta pangenomes (30 common to both) exceeded the threshold of 0.9 indicating they are intolerant to change (Supplementary Data 1). Separately, five and three genes that overlapped with bubbles in hifiasm and Shasta pangenomes are listed in the OMIA (Online Mendelian Inheritance in Animals) database of 168 genes that cause phenotypic variation in cattle (Supplementary Data 1); for instance, both hifiasm and Shasta pangenomes contained a bubble encompassing a 36 bp deletion in *ACAN* resulting in the in-frame deletion of 12 amino acids in the gaur assemblies. Variants in *ACAN* are associated with variation in stature in cattle[48] and other species[49]. The pangenome also recovered the insertion of an 11 Kb segment on chromosome 23 in all assemblies. This segment encompasses *HSPA1B*[50] which is not annotated in ARS-UCD1.2[51]. This 11 Kb insertion is challenging to identify by inspection of short and long read alignments, mainly appearing as elevated coverage over the 2294 bp segment encompassing *HSPA1A*, indicating a similarly sized duplication, with soft-clipped bases extending on both sides (Supplementary Note).

We also investigated genes in this set previously reported to harbor variants that affect phenotype in cattle. For instance, the coding sequence of *QRICH2* overlapped with multiple bubbles indicating tandem repeats of a 30 bp region of the fifth exon (Fig. 7a). Loss of function alleles in mammalian *QRICH2* orthologs lead to multiple morphological abnormalities of the sperm flagella[52,53]. We find that the fifth exon of *QRICH2*, which contains a variable number of tandem repeats (VNTR), is transcribed in high abundance (>30 transcripts per million) in testes of mature taurine bulls (Supplementary Notes). Inspection of long read alignments confirms an expansion of the coding sequence relative to ARS-UCD1.2 that extends the high molecular weight glutenin subunit of the protein by 10 amino acids in our taurine cattle, 60 amino acids in Nellore, and 50 amino acids in gaur (Fig. 7b, c). This SV is challenging to resolve with short reads (Supplementary Notes). Another example of potential trait-associated SV lies in *TAS2R46*, related to bitter taste receptors and associated with adaptation to dietary habitats[54], which overlapped with a 17 Kb deletion in gaur (Fig. 7d, e). This deletion also spanned ENSBTAG00000001761. A final example is found in *PRDM9*, the only known speciation gene in mammals and known to harbor alleles associated with variation in meiotic recombination within and between Bovinae[55–57], where an SV overlapped with copy number variation in the zinc finger array domain (Fig. 7f, g). The Nellore and gaur assemblies contained one zinc finger less, while the paternal haplotype of OxO carried one more relative to ARS-UCD1.2. The maternal haplotype of OxO contained the same number of zinc fingers as ARS-UCD1.2, supporting the intra- and inter-breed/species variation observed for *PRDM9*.

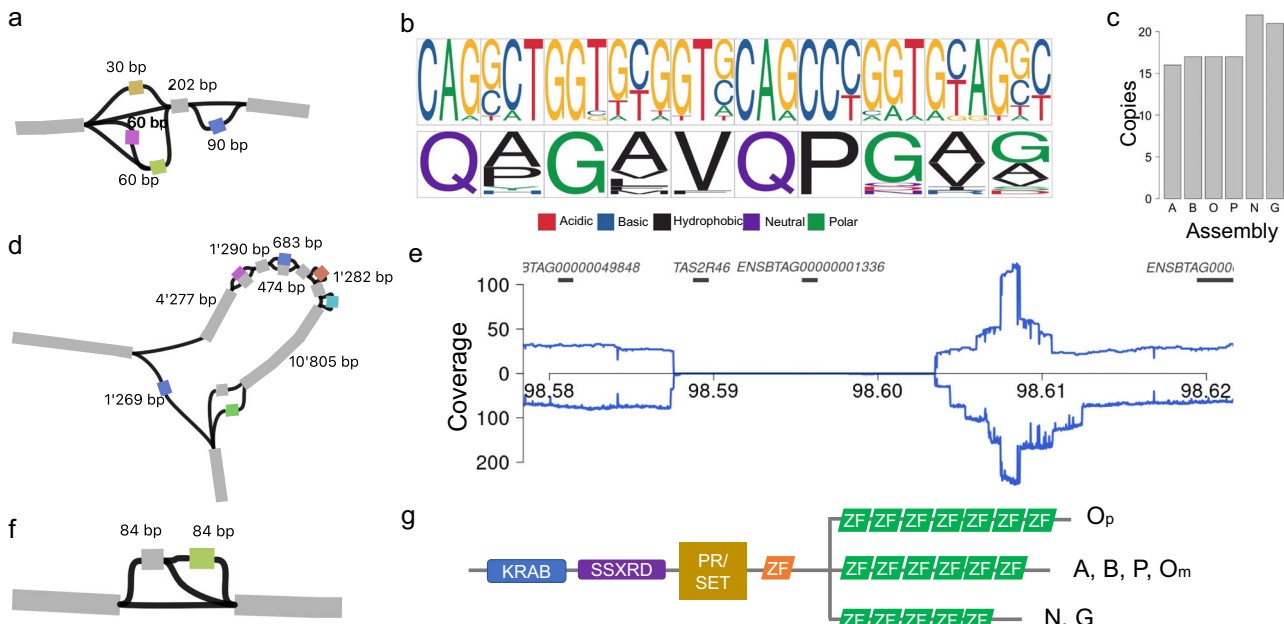

**Fig. 7 Identification of structural variation in coding sequences of *QRICH2*, *TAS2R46* and *PRDM9* through the HiFi-based pangenome. a** Pangenome topology in the fifth exon of bovine *QRICH2* revealed tandem repeats of 30 bp sequence. **b** Nucleotide (upper) and protein (lower) sequence logo plot of the repeat motif. **c** While the ARS-UCD1.2 reference sequence contains 15 copies of the repeat motif, the pangenome revealed 1, 5, and 6 additional copies in the five haplotype-resolved assemblies (A—ARS-UCD1.2, B—Brown Swiss, O—Original Braunvieh, P—Piedmontese, N—Nellore, G—gaur). **d** Representation of a 17 kb deletion on BTA5 encompassing *TAS2R46* and ENSBTAG00000001761. **e** Coverage of binned HiFi (above horizontal line) and ONT (below horizontal line) long read alignments in gaur indicate a large deletion between 98,587,384 and 98,604,401 bp. **f** Pangenome topology at the eleventh exon of *PRDM9* indicating paths with gain and loss of 84 bp sequence. **g** Representation of the domains of *PRDM9* in the haplotype-resolved assemblies including a variable number of zinc fingers (ZF) in the different assemblies, where $O_m$ and $O_p$ are the maternal and paternal haplotypes of the OxO.

## Discussion

Haplotype-resolved assemblies for cattle and related species were constructed using recent sequencing and assembling technologies. Assemblies produced by hifiasm (HiFi) and Shasta (ONT) were substantially more contiguous and correct than those produced by other tools evaluated (Supplementary Table 3) and the current *Bos taurus* reference sequence, which is a haplotype-merged assembly based on older CLR technology. The higher accuracy of HiFi reads was found to be generally more advantageous to quality measures of assembly and increased completeness of centromeric and telomeric regions compared to the longer but higher error ONT reads. HiFi-based assemblers also required less compute and storage resources compared to ONT-based assemblers; producing haplotype-resolved hifiasm assemblies required approximately 600 CPU hours and 200 GB of peak memory usage, while the equivalent Shasta (plus polishing) assemblies took 2200 CPU hours and 750 GB of peak memory usage. Correct-then-assemble approaches like Canu[58] can be practical for smaller genomes[59], but on gigabase-sized mammalian genomes like in Bovinae we observed >20 Tb of peak temporary storage and >25k CPU hours for correcting only 30-fold ONT reads. Even recent reference-guided correction approaches like Ratatosk[60] still needed approximately 15k CPU hours to correct 55-fold ONT reads. Cutting-edge sequencing and bioinformatic improvements[61,62], like the ONT Guppy5 basecaller, will likely assist more efficient assembly, resulting in higher QV and reduced computational load; however, currently the ONT specific requirements might be computationally prohibitive, especially when assembling many samples. HiFi and ONT assemblies can be merged *post hoc*[33] which may improve contiguity[63], although we observed only minor increases (Supplementary Table 8).

The phased assembly graph approach of hifiasm is most efficient with lower heterozygosity samples, where HiFi reads are least sortable by parental haplotype and there is more mutual sequence to exploit, but still functions with highly heterozygous F1s. The interspecies GxP trio binned assemblies were more contiguous (+25% NG50) compared to the trio aware assemblies, but contained several large (>5 Mb) misassemblies which the latter did not (Supplementary Fig. 1b, c). Phasing in both HiFi- and ONT-based assemblies improved as heterozygosity increased and allowed more cleanly resolved haplotypes, which can be beneficial to downstream analyses[64,65]. The ability of hifiasm, and to a lesser extent Shasta with diploid-aware polishing, to assemble phased haplotypes from purebred individuals also avoids ethical and logistical concerns regarding the higher heterozygosity crosses previously targeted[31]. In situations where parental data is unavailable, accurate haplotype phasing is still possible with supplementary data on the offspring[66,67]. Our results show that HiFi reads alone are sufficient for contig-level phasing for higher heterozygosity individuals like the GxP (Supplementary Fig. 7).

The minigraph pangenomes were strongly comparable whether the input assemblies were all HiFi-based, ONT-based, or a mix. The greater completeness observed in hifiasm assemblies is reflected in those pangenomes containing more non-reference sequence, but no notable HiFi or ONT specific biases were observed in the pangenomes. The quality and completeness of the pangenome backbone can have an impact, seen on the incomplete Angus chromosome 28 or the ARS-UCD1.2 chromosomes generally lacking centromeric sequence, but again we found no specific bias between CLR- or ONT-based backbones. These results indicate that optimal minigraph pangenomes would use high-quality, complete genomes as the backbone, like

emerging telomere-to-telomere assemblies[63,68,69]. Alternatively, reference-free approaches that use base-level alignment may circumvent these issues and resolve variation down to single nucleotides[27,70], which may prove crucial for breakpoint resolution and related analyses.

Mutual variation identified through shared paths in the pangenome provides opportunities to study the phylogeny of Bovinae beyond SNPs and indels. The ability to accurately separate and represent paths within SVs also enables pangenome-based GWAS (PWAS)[71], as recently explored in crops[72–74]. We identified multiple SVs overlapping annotated coding sequences in different bovine pangenomes, demonstrating that pangenomes provide a framework to make them amenable to association mapping. Furthermore, some of these SVs (e.g., tandem repeats in *QRICH2*) are inaccessible from short or noisy long read alignments and some (e.g., an insertion of *HSPA1B*) are challenging to resolve even with long read alignments. These cases highlight the benefits of de novo, haplotype-resolved assemblies and pangenome integration which are better able to resolve the variation.

Our work shows that advances in sequencing and algorithms enable computationally feasible haplotype-resolved assembly of 20× HiFi or 60× ONT coverage while retaining more than 90% accuracy of detecting SVs when integrated into pangenomes. Structural variation-based pangenomes built from these assemblies demonstrated significant consensus regardless of sequence platform and heterozygosity of the F1. Given the manageable read input needed, it is feasible to produce in the order of several dozens of haplotype-resolved assemblies for specific breeds of cattle. This effort can be de-centralized as no manual curation is needed to produce assemblies of sufficient quality for SV detection, even from purebred individuals. Due to the low effective population size of most breeds of cattle[75], the resulting pangenomes would capture the most prevalent SVs within breeds, particularly when assemblies were produced from individuals that account for a large portion of the haplotype diversity of the population[76]. More haplotype-resolved assemblies are required to reveal rare SVs[77] and characterize SV prevalence in breeds with large effective population size or a history of admixture[11,78]. Extensive existing short-read sequencing data could then be leveraged to genotype SVs present in the pangenome[24,79] and then impute them into tens of thousands of cattle previously genotyped with microarrays, enabling structural variant analysis at hitherto unattainable scale.

## Methods

**Ethics statement**. The sampling of blood from the NxB and OxO trios was approved by the veterinary office of the Canton of Zurich (animal experimentation permit ZH 200/19). All GxP protocols were approved by the Institutional Animal Care and Use Committee (IACUC) of the University of Nebraska–Lincoln, an AAALAC International Accredited institution (IACUC Project ID 1697). Gaur semen was collected and preserved by Omaha's Henry Doorly Zoo and all protocols were reviewed and approved by their IACUC in 1992.

**Animals**. Cows from the Original Braunvieh (O), Brown Swiss (B), and Piedmontese (P) breeds were inseminated with semen samples from Original Braunvieh, Nellore (N), and gaur (G) sires, respectively. The Original Braunvieh sire/dam and Brown Swiss dam were purebred animals with official entries in the respective Swiss herdbooks. The Piedmontese dam was selected by the American Piedmontese Association as representative of their Piedmontese animals. The Nellore sire is from Brazil, with both parents also recorded as purebred Brazilian Nellore. The gaur sire was a zoo animal. Breed affiliation was examined through a PCA on publicly available cattle whole-genome sequencing samples (Supplementary Fig. 8).

A female (OxO) and a male (NxB) calf were delivered at term. A female fetus (GxP) cross was collected by cesarean section at 118 days of gestation. Blood samples were taken from the calves at approximately 4 weeks of age and dams by trained veterinarians. Lung tissue was taken from the fetus and DNA extracted as described[80]. Semen samples were available for the bulls. High-molecular weight DNA was extracted from blood and semen samples, respectively, using Qiagen's MagAttract HMW DNA Kit as described earlier[18].

**Sequencing**. The genomes of the F1s were sequenced with long reads using PacBio HiFi and ONT. The PromethION 1D Genomic DNA by ligation SQK-LSK110 library prep kit was used for the OxO, NxB, and GxP F1s. The libraries were respectively sequenced on PromethION (R9.4.1) flowcells (nuclease wash was applied to one cell). The OxO and NxB reads were basecalled on Guppy4, while the GxP reads were basecalled with Guppy3.

Paired-end libraries (2 × 150 bp) were produced using parental DNA samples and sequenced on Illumina instruments.

**Genome assembly**. HiFi reads were first filtered with fastp (version 0.21.1)[81], removing reads below 1 Kb length or QV20. Nanopore reads were pre-filtered to a minimum of QV7, with no length restrictions. Short reads were also filtered with fastp, using the "-g" parameter to trim polyG tails.

Both HiFi and ONT reads were trio binned into paternal and maternal haplotypes using (Trio)Canu[31] (version e0d6bb0), using parental short reads and an estimated genome size of 2.7 g.

HiCanu[82] was used with default settings and "genomeSize = 2.7g -pacbio-hifi" to produce a draft set of contigs from the HiFi reads. Only contigs which were not suggested as bubbles (suggestBubble = no) were retained, which improved automated coverage detection when using purge_dups (version 1.2.3)[83]. The final set of "assembled" contigs was obtained after following the default purging pipeline.

The trio aware hifiasm[84] (version 0.15.3-r339) assemblies were produced from HiFi reads with non-default parameters "-a 5 -n 5". The resulting graph information was then used in the trio-mode, with parental k-mers constructed with yak (version 0.1-r62-dirty) specified through the "−1 {paternal} −2 {maternal}" parameters. The haplotype gfa files were then converted to fasta contigs with gfa2fa command of gfatools (version 0.5-r234).

The trio binned hifiasm assemblies were obtained with the same assembly parameters, with the addition of the "–primary" flag to produce a primary and alternate assembly. The primary gfa file was converted to fasta as described above. Only the primary assembly was retained.

Peregrine assemblies[85] (version main:2aefc14+) were produced using the settings "–with-consensus –shimmer-r 3 –best_n_ovlp 8" on either the trio binned or total HiFi data set where appropriate. Only the primary assembly was retained.

Shasta[86] (version 0.7) assemblies were produced using the standard configuration file "Nanopore-Sep2020.conf". Assemblies for the Brown Swiss, Nellore, Piedmontese, and gaur haplotypes used trio binned nanopore reads. The OxO had insufficient coverage to trio bin and then assemble, and so the total F1 nanopore read set was used in assembly. Due to the high coverage for the gaur and Piedmontese haplotypes, the parameter "minReadLength = 30000" was used instead of the default 10000.

All nanopore assemblies except the Original Braunvieh Shasta were polished using PEPPER[87] (version 0.1) and the trio binned nanopore reads. The Original Braunvieh Shasta assembly was polished with PEPPER-DV (version 0.4.1), tagging the intermediate bam file with trio binned read IDs to produce the two Original Braunvieh haplotypes. All assemblies were then further polished using short reads. Since short reads cannot be successfully trio binned, we excluded F1 short reads containing e.g. maternal specific k-mers from the short reads used to polish the paternal haplotype and vice versa using meryl. These short reads were then aligned to the long read polished assembly using BWA-mem2[88,89] and variants were called with DeepVariant[90] (version 1.1). Merfin[91] (commit 1331fa5) was used to filter the variants and then the assembly was polished using bcftools consensus[92] (version 1.12).

Raven[93] (version 1.5.0) assemblies were obtained from nanopore reads with default parameters, except for "-p 0" to disable racon polishing[94]. Assemblies were instead polished with PEPPER for consistency with other ONT assemblers.

Flye[95] (version 2.8.3-b1725) assemblies were constructed with "–genome-size=2.7g –nano-corr" from Ratatosk (version 0.4)[60] error-corrected nanopore reads. The nanopore reads were corrected using a reference-guided approach, taking the haplotype-specific hifiasm assembly as the reference. Ambiguous IUPAC codes were randomly replaced with equal probability of an appropriate nucleotide. The contig set was taken from the pre-scaffolding result of Flye and was not polished due to the pre-corrected reads having sufficient accuracy.

**Scaffolding**. Contigs were scaffolded into chromosomes by the reference-guided approach of RagTag[33] (v2.0.1) to ARS-UCD1.2, using the additional parameters "–mm2-params "-cx asm5" -r -m 1000000".

**Assembly merging**. Hifiasm and Shasta assemblies were merged using RagTag patch, alternatively using either the hifiasm or Shasta assembly as the base assembly, with the same parameters used for RagTag scaffolding. The merged assembly was then scaffolded to ARS-UD1.2 again as described above.

**Quality metrics**. Completeness was assessed with BUSCO (version 5.1.2), using the metaeuk backend (commit 9dee7a7) and odb10 cetartiodactyla database (e96dfc6299c567768085ee9569b6ab15). QV and PG50 were calculated with merqury (version 1.3)[96], with k-mer databases constructed from short reads using meryl (https://github.com/marbl/meryl) (version r953). NG50 was assessed with

calN50.js (https://github.com/lh3/calN50). Repetitive elements were identified by RepeatMasker (http://www.repeatmasker.org) (version 4.1.1) and rmblast (version 2.10.0), using a modified version of the 2018 Repbase database. For whole-genome analysis, the rush job mode was used, while for pangenome region analysis the slow mode was used.

**Coverage downsampling.** Sequencing subsets were made through seqtk (https://github.com/lh3/seqtk) (version 1.3-r115-dirty) with the command "seqtk seq -f {sample}", including the "-A" flag to drop quality scores where appropriate. In the case of repeat trials, the "-s {seed}" flag was set with a randomly generated 64-bit integer to ensure a unique subsampling of data. The reduced coverage assemblies were conducted identically to the full-coverage methods unless explicitly mentioned otherwise.

**Coverage depth estimation.** Sequencing reads were mapped against ARS-UCD1.2 with minimap2 (version 2.19-r1059-dirty)[97]. Coverage depth was determined by megadepth (version 1.1.0c)[98], and averages over 10 Kb windows were estimated with pyBigWig (version 0.3.18)[99].

**Pangenome construction and bubble extraction.** Pangenomes were constructed on a per chromosome basis using minigraph (vesion 0.15-r426)[26], with default parameters and ARS-UCD1.2 as the initial backbone. The selected assemblies were added to the graph in a randomly shuffled order, as "minigraph -xggs <ARS-UCD1.2 > <shuffled input assemblies > ". As input order can affect pangenome construction, this was repeated 20 times with different random orders for each set of inputs (e.g., hifiasm only, Shasta only, mixed, etc.). Pangenomes were visualized with Bandage (version 0.8.1)[100].

Haplotype paths were called with minigraph, using the "–call –xasm" option for each respective assembly, resulting in bed files indicating which node the assembly took through each bubble. Intersection sets for the assemblies across each bubble were created with UpSetPlot (https://github.com/jnothman/UpSetPlot) (version 0.6.0). The intersections were then converted into a condensed distance matrix, where each element in the matrix indicates on how many occasions two assemblies took different paths through the same bubble.

**SNP phylogeny.** Parental short reads were mapped to the ARS-UCD1.2 reference using BWA-mem2. Variants were called using DeepVariant (version 1.1) with the "WGS" model and merged using GLnexus (version 1.3.1)[101]. Phylograms were constructed per chromosome with vcf-kit (version 0.2.9)[102], using the command "vk phylo tree upgma".

**Identifying potential trait-associated SVs.** All entries in the ARS-UCD1.2 annotation (RefSeq release 106) with a "CDS" description were extracted into a bed file. Bubbles from the respective hifiasm or Shasta pangenomes were also extracted into bed files using the command "gfatools bubble". The two bed files were then intersected using bedtools (version 2.30.0)[103] with the intersect command and "-wo" to find SVs that impact coding sequence. We then filtered out SVs where haplotype path information through the bubble was not present for all assemblies, which removed 92 and 77 incomplete or ambiguous associations in hifiasm and Shasta pangenomes.

To assess the significance of the SV-gene overlaps, we collected the lengths for the pangenome bubbles per chromosome and then randomly assigned a new location in the uniform interval [1,chromosome length – bubble length]. These randomly permuted bubble locations were then tested for overlaps with CDS regions as described above. We used a one-sided Wilcoxon signed-rank test, assuming the real bubbles would have fewer overlaps with the CDS regions compared to the permuted bubbles.

The set of overlapped genes was compared against the OMIA database (https://www.omia.org/home/) as well as pLI scores in human orthologues (ftp://ftp.broadinstitute.org/pub/ExAC_release/release1/manuscript_data/forweb_cleaned_exac_r03_march16_z_data.pLI.txt.gz) to produce a table containing all genes of interest overlapping pangenome bubbles. Several specific genes were selected for further investigation based on their presence in OMIA or known existing literature describing trait-associated variants.

**Brown Swiss hifiasm localized reassembly.** HiFi reads in the area of the *GC* bubble were extracted from a bam file of the Brown Swiss trio binned reads aligned to ARS-UCD1.2 using samtools over the interval 6:86900000-87100000. These reads were then assembled using hifiasm as described for standard hifiasm trio binned assemblies and then were used in a local pangenome analysis where required.

**OxO haplotypes.** We used the maternal haplotype of the OxO as the representant assembly for Original Braunvieh (chosen for its more complete X chromosome although it was eventually not used in our analyses) in all analyses where we compared breeds/species. This is the case in Figs. 4, 5a–c, 6, 7a–e, and the Supplementary material except Supplementary Figs. 24, 25. When intra-breed variation

for the Original Braunvieh was relevant, we also included the paternal haplotype of the OxO, used in Figs. 5d, and 7g, f, and the two Supplementary figures listed above.

**Miscellaneous analysis.** Assembly, validation, and pangenome workflows have been implemented using Snakemake[104], and are available online (https://github.com/AnimalGenomicsETH/bovine-assembly). Nucleotide and protein sequence logos were generated using the R package ggseqlogo[105].

**Reporting summary.** Further information on research design is available in the Nature Research Reporting Summary linked to this article.

## Data availability

HiFi reads for the OxO and NxB F1s are available in the ENA database at the study accession PRJEB42335 under sample accession SAMEA7759028 and SAMEA7765441. ONT reads for the OxO and NxB are available in the ENA database at the study accession PRJEB42335 under sample accession SAMEA10017983 and SAMEA10017982. Short reads for the OxO and NxB are available in the ENA database under accession number SAMEA9986200 and SAMEA7589752. Parental short reads are available in the ENA database at SAMEA9986201 & SAMEA9986199 (OxO) and at SAMEA6163185 & SAMEA9533783 (NxB). Long and short read sequencing data for the GxP trio are available in the ENA database at the study accession PRJEB48481 under secondary accessions SAMEA10563833, SAMEA10563834, and SAMEA10563835. The OMIA database is available online (https://www.omia.org/home/) as well as pLI scores in human orthologues (ftp://ftp.broadinstitute.org/pub/ExAC_release/release1/manuscript_data/forweb_cleaned_exac_r03_march16_z_data_pLI.txt.gz). The generated assemblies are available online (https://doi.org/10.5281/ZENODO.5906579).

## Code availability

Code used for generating the results and subsequent analyses in this work can be found at https://github.com/AnimalGenomicsETH/bovine-assembly and on Zenodo at https://doi.org/10.5281/zenodo.6503779[106].

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

## Acknowledgements

The authors thank Dr. Melissa Terranova and Flavio Ferrari (AgroVet-Strickhof) for animal handling and Dr. Sandra Milena Bernal Ulloa for sampling blood. We are thankful for the technical support provided by Dr. Anna Bratus-Neuenschwander from the ETH Zürich technology platform FGCZ (https://fgcz.ch) for sequencing and DNA fragment analysis. The results reported here were made possible with resources provided by the USDA shared compute cluster (Ceres) as part of the ARS SciNet initiative. We thank the USMARC Core Facility staff for outstanding technical assistance. We also thank B. Lee, J. Carlson, K. McClure, H. Clark, H. Sadd, M. Sadd, and B. Shuck for outstanding technical support. We thank the North American Piedmontese Association for their enthusiastic support and assistance in producing gaur interspecies crosses, and Dr. Jason Herrick and Omaha's Henry Doorly Zoo for the gaur photograph and semen. Mention of trade names or commercial products in this publication is solely for the purpose of providing specific information and does not imply recommendation or endorsement by the U.S. Department of Agriculture. USDA is an equal opportunity provider and employer. H.P. was financially supported from the Federal Office for Agriculture (FOAG), Bern, Switzerland, and the Swiss National Science Foundation (SNSF, project number 185229). The United States Department of Agriculture, Agricultural Research Service appropriated projects (3040-31000-100-00D) to T.P.L.S. and L.A.K., and (5438-32000-034-00D) to M.P.H.; B.L.V.L. was financially supported by the University of Nebraska Great Plains Veterinary Educational Center project (2162390003) and the Nebraska Beef Industry Endowment (2662390323001). The reproductive and animal husbandry portions of the project were supported by the North American Piedmontese Association.

## Author contributions

A.S.L., D.M.B., T.P.L.S., B.D.R., and H.P. jointly conceived the study. Z.F., M.P.H., B.L.V.L., C.H., H.B., and K.L.K. contributed to sample collection and sequencing. A.S.L. performed the genome assembly, validation, and pangenome construction. A.S.L. and D.C. performed the pangenome analysis. A.S.L. and H.P. performed the trait-associated and SV-coding overlap analyses. A.S.L. and H.P. wrote the initial manuscript, and A.S.L., D.M.B., T.P.L.S., B.D.R., and H.P. revised the manuscript.

## Competing interests

The authors declare no competing interests.
