## [Peer Review File · Nature Communications]

Title: Structural variant-based pangenome construction has low sensitivity to variability of haplotype-resolved bovine assembliesREVIEWER COMMENTS

Reviewer #1 (Remarks to the Author):

Leonard et al. present an interesting analysis that is really broken down into two parts. An assessment of different strategies of building haplotype resolved assemblies using parental information and a subsequent analysis of SVs in these genomes. It likely reflects my personal biases but I think the first part is stronger, with the analysis of SVs largely restricted to a couple of anecdotes about known existing SVs or SVs that may fall in genes with a relevant downstream function. Even the total list of effected genes arent listed as far as I can tell and this bit could arguably be dug into a bit more as I describe below. My comments in no particular order are:

Abstract, line 25. "Ten haplotype-resolved assemblies of three bovine trios" I think as written this is potentially confusing. The trios weren't assembled, just the offspring. More accurately the authors appear to generate 12 haplotype-resolved assemblies of three animals, using trio-binning. Two assemblies per haplotype, having generated each from ONT and HiFi data separately. They then say "Variation between the paternal and maternal haplotypes of the OxO, reflecting within-breed diversity, was sufficient to generate haplotype-resolved assemblies, although only the maternal haplotype was used to represent the Original Braunvieh breed in subsequent analyses except where explicitly stated." So they decided to drop the two Original Braunvieh paternal haplotypes. But if these were successfully assembled why are they not presented in e.g. Table 1 etc? Why was the paternal genome chosen to be dropped and not the maternal one given are mostly focusing on the autosomes? They then go on to use both haplotypes downstream making its absence in table 1 more confusing. So I think things need to be presented more clearly around these points.

This paper is really split into two parts, the first is an assessment of different strategies of generating assemblies from different types of trios. The second is an assessment of SVs in these genomes. From the title the authors focus seems to be on the second part and calling SVs. If this is the case why did they not try to generate hybrid assemblies through combining both the HiFi and ONT data? This is expected to give better assemblies than either alone and improve the downstream analyses.

Willing to be persuaded otherwise but I am not convinced by the use of the term "Bovine pangenome" in the title when are talking about just five genomes. For example, with one Nellore genome the authors surely wouldn't suggest they have the pan-genome of this breed, so therefore they cant really suggest they have it for Bovine as a whole. Although can talk about the pangenome of a set of animals that is different to a Bovine pangenome.

Be good to see the haplotypes on a PCA alongside "pure" representatives of key divergent breeds (or at least of the parents). It is always nice to confirm that the animals cluster where they should, especially for breeds such as Nellore which can often be introgressed and not pure indicus. Also may indicate if any haplotype mix-up.

Figure 1e. How is read separability measured? Doesn't appear to be described anywhere. If its just the proportion of reads that are assigned to a parent then how much error is there in this measurement? I assume sometimes a read can be assigned to the wrong parent? Also be good to indicate the cross each pair of points refers to in Fig 1e e.g. by using same colour scheme as Fig 1d

Line 139 shouldn't begin with Table 1 I believe.

Am I right in thinking Gaur do not have the same karyotype as cattle? If so the authors don't seem to cover this. For example, how is this effecting the assembly of the haplotypes of its offspring? Will it potentially not be missing chromosomes (as the gaur chromosome number is lower than cattle's)? Was the Gaur also scaffolded versus the Hereford with RaGOO as suggested on line 123? If so is this not potentially going to cause problems downstream if the karyotype is different?

The authors are highlighting how the trio based assembly of the offspring is effective at generating haplotype-resolved assemblies. A minor point but as trio-based assembly has comparative difficulties with regards to getting the requires samples it maybe would have been nice to also see how much better the trio based phasing turns out to be over approaches that don't require parental information e.g. phasebook.

Did the authors do anything in their downstream analyses regarding the low quality regions identified by assemblers such as hifiasm? Polish them or take them into account in downstream results?

I don't believe "Autosomal size" and "Genome size" in the legend of Figure 3 should be capitalized.

"Pangenomes for each autosome were constructed from all hifiasm assemblies, all Shasta assemblies, or random mixtures of the two". I may have missed it but the authors don't seem to say how many times for each? Figure 4B has error bars (though of what type isn't described) so I assume they have constructed the pangenomes multiple times, changing the order in which each genome is added. But doesn't say how many times? The description of how the pangenomes are made is just two sentences (679-682) that I don't think gives enough information to understand this. Generally I would say the methods section is too sparse.

"counting the number of mutually exclusive bubbles any two assemblies have to construct a condensed distance matrix" be good to have more details on precisely how this was done. Again, the methods don't seem to provide any further details.

I got lost by the colours in Figure 5A and B. Figure 5B says the colours come from 5A. But 5A doesn't define what the colours mean. Is it just indicating the three possible orientations? If so Fig 5B seems to be indicating how many autosomes carry each orientation for each graph, but how does this differ depending on the order of graph construction for example? This may be related to my point above where I am assuming the graphs were constructed multiple times, changing the order of adding genomes (which I think the authors state did in the methods) but if this is the case can this number not

change? Or is it fixed and independent of order of graph construction? Surely the mix category cant have just been done once so these should have some form of error bars? Or at least it is made clear this numbers are stable.

“The reported CNV was only observed in the Shasta assemblies of Brown Swiss and Original Braunvieh cattle (Figure 6a,b)” is there not also an increase in read depth in the hifiasm assembly plots for these breeds? Or am I misunderstanding what the authors mean here?

“Of these, 808 and 845 are bubbles with haplotype path information for each assembly, and so are amenable to association investigations. Several of these genes are listed in the OMIA (Online Mendelian Inheritance in Animals) database as they harbour alleles causing phenotypic variation in cattle”. Is this more or less than you would expect by chance? Can you predict the coding consequence of the changes e.g. with VEP? Are the SVs in genes intolerant to change e.g. with high or low pLI scores of their human orthologues? I would hypothesis these genes are disproportionately those where a change is tolerated e.g. olfactory receptors etc but the author don't seem to list which genes they are anywhere?

Line 648: “Contigs were scaffolded into chromosomes by the reference-guided approach of RagTag (Alonge et al., 2019)”. That is the reference for RaGOO which RagTag supersedes. I believe the correct reference for RagTag is Alonge, Michael, et al. "Automated assembly scaffolding elevates a new tomato system for high-throughput genome editing." *bioRxiv* (2021). <https://doi.org/10.1101/2021.11.18.469135>. Also needs correcting in results section.

“Combined, these results support that cataloguing most within-breed structural variation is a tractable goal through de novo long read assembly of 50 samples per breed, sufficient to represent the effective population size.” I don't see how came up with this number? Surely depends on various factors e.g. the breed being studied, the diversity within the breed, the diversity between the animals selected to be sequenced, the minimum allele frequency of the SVs you want to capture. I think unless can better justify this statement should be removed.

Minigraph is only one way of constructing pangenomes and it may be worth adding a sentence or two to the discussion regarding the other possible methods that could have been used and their advantages and disadvantages. For example although minigraph has advantages such as it is fast it does have limitations versus reference-free approaches such as cactus.

Availability of data – are the authors not going to make the assembled genomes available somewhere?

Reviewer #2 (Remarks to the Author):

Review – Leonard et al., Bovine pangenome reveals trait-associated structural variation from diverse

assemblies

This manuscript describes the thorough investigation of 5 de novo assemblies using 2 different third generation sequencers and several different methods to create haplotype-resolved assemblies. The authors compared the performance of phasing between different levels of heterozygosity in the F1. They explored the minimal sequencing depth needed to reach given standards, by down-sampling the sequence data. Next, they created pan-genomes in different combinations and with different reference genomes as backbone, to compare phylogeny and to identify genes located in structural variation bubbles that may be trait associated.

It is a well written and structured manuscript, my compliments to the authors to present all the results of the many different methods applied in a concise way. However, I have 2 general comments and a number of line-by-line comments.

General comments

1) I am missing a conclusion at the end of the discussion.

2) For the trait associated SV results, it is not very clear which steps were taken to prioritize the presented results and it is not clear what portion of the detected SVs in the pan-genomes is (known to be) trait associated. The title puts a lot of emphasis on this part, and I missed a sort of overview from which the examples were picked to explain in more detail.

The authors start off with 2 specific SVs, but it is not clear why these were chosen to report on in detail. They are certainly of interest, but please guide the reader. Thereafter, ~900 genes are mentioned of which several are in OMIA, but it is not clear how many. Also, how the set of genes in the last paragraph (starting at L453) were prioritized is not given. Were there cross references with other databased besides OMIA? Taken these comments together it would be nice to get a better feeling of how many SVs were detected, and how many are (known/likely to be) trait associated, in addition to the ones that are described in detail. Also, I suggest to add (to Material & methods) a section to explain how you declared your detected SVs as trait associated.

Line by line comments

L61-63 I suggest to reformulate this sentence, because larger SV, like CNV, can be and have been detected using short read sequencing in cattle (e.g. Mei et al. 2020; Butty et al., 2020; Kommadath et al., 2019). Of course repetitive regions and reference genome quality hamper SV detection using short reads and needs to be mentioned. Moreover, given the title, the SV topic is rather limited covered in the introduction. It may be highlighted that third generation technology can be beneficial in complex regions.

L113 add '(SR)' behind short reads to make clear this is what the abbreviation in the Figure 1 stands for.

L139 remove 'Table 1' at start of sentence

L170-173 It is not clear to me how you can separate the 2 facts, lower coverage and lower heterozygosity compared to the other F1's, in this matter. Could they not both have an effect on PG50?

L175-189 The generated HiFi assemblies are especially longer due to the centromeres and telomeres, I

was wondering how much longer it is excluding centromere and telomere regions, as the remaining additional length is probably more interesting from a functional perspective, but may be limited.

L201 not sure what is meant with routinely in this context

L213 CLR not yet defined, please add

L223-234 It is a pity that PG50 is not included in Figure 3, that could back up your comment in L170-173.

If a lower coverage in this more heterozygote cross would also show drop in PG50, it becomes more plausible that indeed the lower heterozygosity did not play a role in PG50 for OxO.

L383 Pausch et al. detected the QTL, however they didn't report the presence of a CNV, so either rephrase the sentence or remove the reference.

L386-387 also here the relevance of the reference is not clear to me, please adjust

L396-398 Did you have a look at the parental SR data, that may help in resolving the copy number

L441 Please indicate how many are listed in OMIA, as several is rather vague

L453 How were these 'further putatively trait-associated regions' detected, did they also harbour genes in OMIA or from some other database?

L457-459 Any reference to substantiate this finding (which doesn't seem to be part of this study)?

L532 T2T not defined

L545-546 what exactly is then the benefit if they are still difficult to resolve with de novo haplotype resolved assemblies and pangenomes?

L553 this 50 seems a rather arbitrary number or is it based on effective population size of cattle breeds?

L556-557 just a comment: LD between SV and SNP is low (as you indicate in L363) and hence SV may not be accurately imputed into array genotypes, but that needs to be investigated.

L616 Please add what Peregrine was used for.

L625 OBV? Think it was defined as O, please correct throughout manuscript.

L635-637 does this mean you have done both? Which one is reported/used in this study?

L656 Maybe add here that with quality you refer to NG50 and with phasing to PG50 for clarification

REVIEWER COMMENTS

Reviewer #1 (Remarks to the Author):

Leonard et al. present an interesting analysis that is really broken down into two parts. An assessment of different strategies of building haplotype resolved assemblies using parental information and a subsequent analysis of SVs in these genomes. It likely reflects my personal biases but I think the first part is stronger, with the analysis of SVs largely restricted to a couple of anecdotes about known existing SVs or SVs that may fall in genes with a relevant downstream function. Even the total list of effected genes arent listed as far as I can tell and this bit could arguably be dug into a bit more as I describe below.

We thank the reviewer for their time to review our manuscript and for raising valuable comments and suggestions which improved our manuscript. The intent of the work is to examine the effect of including genome assemblies of variable quality and assembly strategy on pangenome construction and application. The reviewer states the two parts of the manuscript are (i) assessing genome assembly strategies and (ii) SV analysis. However, the two parts of the manuscript are better characterized as (i) pangenome graph construction from disparate assembly types/underlying data and quality comparisons of these graphs followed by (ii) comparison of these pangenomes for application to SV detection. The assessment of assemblies prior to pangenome graph construction was a necessary step in the overall analysis but does not constitute a main goal of the work, as comparisons of sequencing technologies or trio versus single individual assembly has previously been reported. The analysis of SV is principally a means to evaluate the utility of the various pangenome graphs constructed, but as the overall value of pangenome graphs versus single reference assemblies has not been firmly established, particularly in bovine, we determined that the utility for SV detection of pangenomes constructed from even this relatively small sample of animals was worthwhile to include in the report. In order to clarify the motivation of our analyses and guide the reader through the different sections of the manuscript, we added introductory sentences at various sections (lines 119, 334, and 387).

My comments in no particular order are:

Abstract, line 25. "Ten haplotype-resolved assemblies of three bovine trios" I think as written this is potentially confusing. The trios weren't assembled, just the offspring. More accurately the authors appear to generate 12 haplotype-resolved assemblies of three animals, using trio-binning. Two assemblies per haplotype, having generated each from ONT and HiFi data separately. They then say "Variation between the paternal and maternal haplotypes of the OxO, reflecting within-breed diversity, was sufficient to generate haplotype-resolved assemblies, although only the maternal haplotype was used to represent the Original Braunvieh breed in subsequent analyses except where explicitly stated." So they decided to drop the two Original Braunvieh paternal haplotypes. But if

these were successfully assembled why are they not presented in e.g. Table 1 etc? Why was the paternal genome chosen to be dropped and not the maternal one given are mostly focusing on the autosomes? They then go on to use both haplotypes downstream making its absence in table 1 more confusing. So I think things need to be presented more clearly around these points.

We have removed the references to “Ten” haplotype-resolved assemblies in the introduction and discussion, and have added the paternal OxO haplotype to Table 1. We have also clarified when we use which haplotypes in the main text (lines 129-132):

“resulting in six haplotype-resolved assemblies, five of which represent novel breed/species. For breed/species-level analysis, we only use the maternal haplotype of the OxO by choice, while analyses displaying intra-breed variation use both haplotypes (methods).”

As well as explicitly stating in the methods (lines 772-779):

“We used the maternal haplotype of the OxO as the representant assembly for Original Braunvieh (chosen for its more complete X chromosome although it was eventually not used in our analyses) in all analyses where we compared breed/species. This is the case in Figure 4, Figure 5a-c, Figure 6, Figure 7a-e, and the Supplementary material except Figure_SN_1_16 and Figure_SN_1_17. When intra-breed variation for the Original Braunvieh was relevant, we also included the paternal haplotype of the OxO, used in Figure 5d, Figure 7g,f, and the two Supplementary figures listed above.”

This paper is really split into two parts, the first is an assessment of different strategies of generating assemblies from different types of trios. The second is an assessment of SVs in these genomes. From the title the authors focus seems to be on the second part and calling SVs. If this is the case why did they not try to generate hybrid assemblies through combining both the HiFi and ONT data? This is expected to give better assemblies than either alone and improve the downstream analyses.

The analyses in our manuscript were designed in a way to investigate if disparate assemblies generated from modern sequencing and assembly approaches can be integrated into pangenomes without compromising SV detection from the pangenome. Specifically, we investigate SV detection from assemblies produced from current assemblers that don't require manual curation efforts or additional data types (such as Hi-C). Large manual curation efforts would likely prevent assembly and integration of many assemblies (discussion). Moreover, we are not aware of any assemblers which can natively use both ONT and HiFi (see discussions at <https://github.com/chhylp123/hifiasm/issues/19>, <https://github.com/fenderglass/Flye/issues/312>, <https://canu.readthedocs.io/en/latest/quick-start.html#assembling-with-multiple-technologies-and-multiple-files>). Approaches which do combine ONT and HiFi data generally involve substantial manual steps to create the assembly (Nurk et al., 2021) or require post-

assembly curation (van Rengs et al., 2021; B. Wang et al., 2021). We have added additional discussion on combining ONT and HiFi data to clarify this point (lines 521-524).

“Currently there are no approaches to co-assembling HiFi and ONT data due to different read properties; however, HiFi and ONT assemblies can be merged *post hoc* (Alonge et al., 2021) which may improve contiguity (van Rengs et al., 2021), although we observed only minor increases (Supplementary Table 9).”

We do not observe any substantial improvement in contiguity when combining the hifiasm and Shasta assemblies (presented as new Supplementary Table 9 in the revised manuscript), where the merged contiguity is rarely better than the maximum of hifiasm and Shasta. This minor gain in continuity is likely due to the already high contig NG50 of the sole-data source assemblies to begin with. Furthermore, the incorporation of contigs assembled from different sequencing technologies may inadvertently add new assembly errors that offset any potential benefits gained by these small increases in continuity. Identification and classification of errors resulting from assembly merger will require additional efforts that are beyond the scope of our current work. We also show that reduced contiguity in the lower-coverage assemblies does not substantially impact SV discovery through pangenomes, and so maximising contiguity is not strictly necessary.

Supplementary Table 9. NG50 for merged hifiasm and Shasta assemblies. Assemblies were merged with RagTag for each assembly pair separately, either merging the Shasta assembly into the hifiasm assembly (Merge [HiFi-backed]) or vice versa. Contig NG50 values are given in Mb.

	Breeds/species					
	O (pat)	O (mat)	N	B	G	P
HiFi	56.0	47.0	94.4	86.7	73.5	52.0
ONT	71.6	71.7	68.5	64.0	68.1	82.8
Merge (HiFi-backed)	62.4	65.6	96.4	86.7	73.5	65.2
Merge (ONT-backed)	72.9	72.9	68.5	64.0	73.7	84.1

Willing to be persuaded otherwise but I am not convinced by the use of the term “Bovine pangenome” in the title when are talking about just five genomes. For example, with one Nellore genome the authors surely wouldn’t suggest they have the pan-genome of this breed, so therefore they cant really suggest they have it for Bovine as a whole. Although can talk about the pangenome of a set of animals that is different to a Bovine pangenome.

We agree with the reviewer that these five breed/species do not representant all bovine diversity, or even all diversity for the represented breed/species. We believe the term pangenome is reasonably interpreted as a collection of genomes integrated into a single structure, as used in recent literature (Li et al., 2022; K. Wang et al., 2021; Zanini et al., 2021), as opposed to encompassing all diversity. Although a term like “multi-assembly graph

genome” may be more specific, we believe that pangenome is a more accessible term to a wider audience. However, we have suggested a change of title from

Bovine pangenome reveals trait-associated structural variation from diverse assembly inputs

to

Bovine pangenome from twelve haplotype-resolved *de novo* assemblies of disparate origins reveals trait-associated structural variation

which makes explicit this pangenome is a sample of bovine diversity coming from a set of *de novo* assemblies.

Be good to see the haplotypes on a PCA alongside “pure” representatives of key divergent breeds (or at least of the parents). It is always nice to confirm that the animals cluster where they should, especially for breeds such as Nellore which can often be introgressed and not pure indicus. Also may indicate if any haplotype mix-up.

We have expanded the methods section on “Animals” to indicate the origin of the animals used in the study, as well as including a PCA of related samples in Supplementary Figure 8 showing the animals cluster where expected (lines 591-596).

“The Original Braunvieh sire/dam and Brown Swiss dam were purebred animals with official entries in the respective Swiss herdbook populations. The Piedmontese dam was selected by the American Piedmontese Association as representative of their Piedmontese animals. The Nellore sire is from Brazil, with both parents also recorded as purebred Brazilian Nellore. The gaur sire was a zoo animal. Breed affiliation was examined through a PCA on publicly available cattle whole-genome sequencing samples (now included as Supplementary Figure 8).”

Supplementary Figure 1. PCA of taurine (Holstein, Piedmontese, Brown Swiss, Original Braunvieh), indicine (Nellore) and indicine x taurine (NxB) crosses. Different colors separate individuals by breed. Red framed symbols highlight the purebred animals from the three trios. The PCA was performed for all samples (a) and for the taurine-only (N=41) samples (b). Multi-sample variant calling was done on publicly available whole-genome sequencing data of 10 Brown Swiss (SAMEA5415485, SAMEA7573645, SAMEA7573582, SAMEA7573592, SAMEA7573599, SAMEA5714971, SAMEA6163191, SAMEA7573585, SAMEA6163179, SAMEA6163195), 10 Original Braunvieh (SAMEA4827668, SAMEA5059753, SAMEA5159850, SAMEA6272091, SAMEA4827645, SAMEA4827672, SAMEA6272103, SAMEA6272093, SAMEA5564728, SAMEA5059754), 10 Holstein (SAMEA6528904, SAMEA6528905, SAMEA6528906, SAMEA6528907, SAMEA6528908, SAMEA6528909, SAMEA7015110, SAMEA7015112, SAMEA7015113, SAMEA7015115), 6 Piedmontese (SAMEA5159836, SAMEA19324918, SAMEA33001918, SAMN02941216, SAMN02941219, SAMN02941218) and 5 Nellore (SAMN10486398, SAMN10486400, SAMN10486401, SAMN10486399, SAMN03387027) samples together with short sequencing reads from the OxO and NxB trio as well as the dam of the GxP trio. There were 35.4 million and 20.1 million polymorphic sites along the autosomes used to construct a genomic relationship matrix for the taurine/indicine and taurine-only breeds, respectively.

Figure 1e. How is read separability measured? Doesn't appear to be described anywhere. If its just the proportion of reads that are assigned to a parent then how much error is there in this measurement? I assume sometimes a read can be assigned to the wrong parent? Also be good to indicate the cross each pair of points refers to in Fig 1e e.g. by using same colour scheme as Fig 1d

We rephrased this to make the definition that separability is indeed the proportion of reads assigned to maternal/paternal bins (lines 102-104).

Previous:

“The success of parent-of-origin assignment improved significantly from 81.1% to 99.9% with increasing heterozygosity for HiFi reads,”

New:

“Separability, the proportion of reads assigned to a parent-of-origin bin, improved significantly from 81.1% to 99.9% with increasing heterozygosity for HiFi reads,”

Reads are assigned to maternal/paternal/unknown bins based on the support from a filtered set of k-mers (as described in Koren et al., 2018). As such, there is not a straightforward way to estimate error rates in binning, as reads are assigned based on the closest matching parent and there is no other “truth” set to compare against. However, we are confident the binnings are overwhelmingly accurate, as observed through the reported high quality phasing and low quantity of phase switches.

Line 139 shouldn't begin with Table 1 I believe.

Corrected.

Am I right in thinking Gaur do not have the same karyotype as cattle? If so the authors don't seem to cover this. For example, how is this effecting the assembly of the haplotypes of its offspring? Will it potentially not be missing chromosomes (as the gaur chromosome number is lower than cattle's)? Was the Gaur also scaffolded versus the Hereford with RaGOO as suggested on line 123? If so is this not potentially going to cause problems downstream if the karyotype is different?

We thank the reviewer for raising this. The (Indian) Gaur sire is expected to have a $2n=58$ karyotype (Mamat-Hamidi et al., 2012), compared to the typical cattle $2n=60$, resulting from a centromeric fusion of chromosomes 2 and 28. No assemblers were able to assemble through the possible centromeric fusion, but the strong synteny between the gaur and cattle genomes allows the reference-guided scaffolding to ARS-UCD1.2 to work as intended, correctly scaffolding Gaur2 to Cattle2, and Gaur28 to Cattle28. Our downstream analysis did not involve karyotypes explicitly, and so scaffolding to ARS-UCD1.2 rather than a true gaur karyotype also simplified the per-autosome pangenomes and related analyses. This has been clarified in the revised manuscript (lines 124-127), and we have added a new panel (panel a) to what was Supplementary Figure 7 and is now Supplementary Figure 1.

“The gaur has a centromeric fusion of chromosomes 2 and 28 resulting in a different karyotype than that of domestic cattle (Mamat-Hamidi et al., 2012), but this scaffolding

approach was still successful for the paternal haplotype of the GxP (Supplementary Figure 1a).”

Supplementary Figure 2. Alignment plots of hifiasm gaur assemblies and ARS-UCD1.2. a) Dot plot of chromosomes 2 and 28 in gaur (left) and ARS-UCD1.2 (bottom). Since the gaur assembly did not span the centromeric fusion, the scaffolding to chromosomes 2 and 28 behave normally.

The authors are highlighting how the trio based assembly of the offspring is effective at generating haplotype-resolved assemblies. A minor point but as trio-based assembly has comparative difficulties with regards to getting the requires samples it maybe would have been nice to also see how much better the trio based phasing turns out to be over approaches that don't require parental information e.g. phasebook.

Getting parental information can be limiting, so we have added additional discussion on haplotype-resolved assemblies created without additional data or with purely offspring/individual data like HiC, with the new Supplementary Figure 7 indicating the ability to resolve haplotypes without parental information and supporting references (lines 536-540).

“In situations where parental data is unavailable, accurate haplotype phasing is still possible with supplementary data on the offspring (Cheng et al., 2021; Porubsky et al., 2020). Our results show that HiFi reads alone are sufficient for contig-level phasing for higher heterozygosity individuals like the GxP (Supplementary Figure 7).”

Supplementary Figure 3. Merqury phasing blob plots for OxO, NxB, and GxP hifiasm assemblies using parental information (trio) or using a local phasing algorithm in hifiasm using only the F1 HiFi reads (dual). Blobs represent individual contigs, and the axes represent how many specific dam/sire haplotype k-mers (hapmers) the contig contained. Hap1 and hap2 refers to the paternal and maternal haplotypes respectively. Phasing is highly accurate when using parental information (top). Without parental information, increasing heterozygosity allows improved phasing. The mean shortest Euclidian distance for each blob to the diagonal (greater distance indicates a haplotype is more strongly maternal or paternal) increases for the OxO, NxB, and GxP from 3204.5 to 36322.6 and then 138001.4. Hifiasm cannot infer if a contig belongs to the maternal or paternal assembly since there is no parental information but, given enough heterozygosity (e.g., the GxP), is able to consistently phase haplotypes to the contig level.

We thank the reviewer for the reference to phasebook (Luo et al., 2021), however, such a tool reported 20x runtime compared to hifiasm and has N50 values on the order of Kbs rather than Mbs, making it unsuitable for our study on highly contiguous assemblies. Similar assemble-phase-assemble approaches like DipAsm (C.-S. Chin et al., 2020) likewise have increased runtime (having to assemble twice plus phasing) without assembly metric improvements over approaches using offspring-only HiC (Cheng et al., 2021) or strand-seq (Porubsky et al., 2020). As such we did not consider them.

Did the authors do anything in their downstream analyses regarding the low quality regions identified by assemblers such as hifiasm? Polish them or take them into account in downstream results?

We did not pursue the low quality regions for downstream analysis, as they are only identified by a few assemblers and with different and loosely defined criteria. In the case of hifiasm, the low quality regions accounted for 30 ± 9 Mb of sequence per genome, with an average of 55% of that sequence in unplaced scaffolds. These low quality regions also tend to occur at contig ends (35% of the low quality bases occur in the first or last 10% of contig length) and are generally highly repetitive (81.2% masked, 65.4% are satellites). Due to the difficulty in resolving these regions with accurate long reads, polishing with short reads is unlikely to improve them, and potentially could cause additional errors. Furthermore, there was no overlap between low quality regions and bubbles identified in the hifiasm pangenome, indicating these regions don't compromise our downstream SV analysis.

I don't believe "Autosomal size" and "Genome size" in the legend of Figure 3 should be capitalized.

Corrected.

"Pangenomes for each autosome were constructed from all hifiasm assemblies, all Shasta assemblies, or random mixtures of the two". I may have missed it but the authors don't seem to say how many times for each? Figure 4B has error bars (though of what type isn't described) so I assume they have constructed the pangenomes multiple times, changing the order in which each genome is added. But doesn't say how many times? The description of how the pangenomes are made is just two sentences (679-682) that I don't think gives enough information to understand this. Generally I would say the methods section is too sparse.

The last sentence in the caption of figure 4 is "All points reflect the mean over 20 stochastic pangenome constructions, and error bars indicate the 95% confidence interval." For clarity we have expanded the methods section on how we constructed pangenomes and how many times we repeated construction (lines 718-722).

"The selected assemblies were added to the graph in a randomly shuffled order, as "minigraph -xggs <ARS-UCD1.2> <shuffled input assemblies>". As input order can affect pangenome construction, this was repeated 20 times with different random orders for each set of inputs (e.g., hifiasm only, Shasta only, mixed, etc.)."

"counting the number of mutually exclusive bubbles any two assemblies have to construct a condensed distance matrix" be good to have more details on precisely how this was done. Again, the methods don't seem to provide any further details.

We've expanded the methods under the section "Pangenome construction and bubble extraction" to include additional detail on pangenome construction and the steps taken to convert pangenome bubbles into the condensed distance matrix (lines 725-731).

“Haplotype paths were called with minigraph, using the “--call --xasm” option for each respective assembly, resulting in bed files indicating which node the assembly took through each bubble. Intersection sets for the assemblies across each bubble were created with UpSetPlot (<https://github.com/jnothman/UpSetPlot>) (version 0.6.0). The intersections were then converted into a condensed distance matrix, where each element in the matrix indicates on how many occasions two assemblies took different paths through the same bubble.”

I got lost by the colours in Figure 5A and B. Figure 5B says the colours come from 5A. But 5A doesn't define what the colours mean. Is it just indicating the three possible orientations? If so Fig 5B seems to be indicating how many autosomes carry each orientation for each graph, but how does this differ depending on the order of graph construction for example? This may be related to my point above where I am assuming the graphs were constructed multiple times, changing the order of adding genomes (which I think the authors state did in the methods) but if this is the case can this number not change? Or is it fixed and independent of order of graph construction? Surely the mix category cant have just been done once so these should have some form of error bars? Or at least it is made clear this numbers are stable.

For clarity, we added the following sentence to the Figure 5 caption (line 350).

“The colored boxes represent the three possible orderings of the taurine cattle”

We have also changed Figure 5b to be a box plot rather than a bar plot, to both better indicate the repeated pangenome construction as well as the underlying variation. We have added the following text to the caption of Figure 5 to this effect (lines 351-354).

“Box plots of 20 randomly constructed pangenomes with either all hifiasm, all Shasta, or mixing hifiasm and Shasta assemblies, as well as the low coverage equivalents show good agreement on autosomes displaying a specific topology (color from panel a). The box plots represent the median (center line), first and third quartile (box bottom and top), and 1.5x the interquartile range (whiskers). Outliers beyond this range are marked by diamond markers.”

“The reported CNV was only observed in the Shasta assemblies of Brown Swiss and Original Braunvieh cattle (Figure 6a,b)” is there not also an increase in read depth in the hifiasm assembly plots for these breeds? Or am I misunderstanding what the authors mean here?

We have restructured this section to clarify the presentation of our results (starting on line 387). Furthermore, we have updated Figure 6a to include an updated hifiasm pangenome which highlights the difference in copy number between HiFi and ONT assemblies, made explicit in the caption of Figure 6 (line 412-415).

“There presently is no “truth set” of bovine structural variants appearing in the haplotypes represented to benchmark the success of each pangenome graph. Therefore, we selected several previously identified SVs known to impact phenotype which were present in our pangenomes for further investigation. One case is a multi-allelic copy number variation (CNV) at 86.96 Mb on BTA6 encompassing an enhancer of the group-specific component (GC) gene. This CNV has pleiotropic effects on mastitis resistance and other dairy traits, and segregates in different breeds of cattle (Lee et al., 2021; Olsen et al., 2016), although the prevalence of the CNV has not yet been determined for the breeds/species included in our bovine pangenome. All examined assemblers predicted duplication of a 12 Kb segment in Brown Swiss and Original Braunvieh (Figure 6a,b, Supplementary Figure 5), although hifiasm required reassembling a small subset of reads as the duplication was missing in the original assembly (methods). The duplicated segments contained various repetitive elements (Figure 6c) matching those previously reported (Lee et al., 2021). There was poor consensus in copy number among the assemblies (Supplementary Figure 5), indicating that *de novo* assembly will still benefit from increased read length and accuracy. Inspection of coverage data retrieved from HiFi- and ONT-binned long read alignments against the ARS-UCD1.2 reference confirmed that the gaur, Piedmontese, and Nellore haplotypes have no additional copies, while the Brown Swiss and Original Braunvieh haplotypes harbor between 2 and 4 additional copies of the 12 Kb CNV segment (Figure 6d, Supplementary Notes). Coverages derived from the NxB and OxO F1 short read alignments are consistent with these values but are unable to resolve the haplotype-specific copy number disagreement observed between HiFi and ONT coverage, as well as between alignments and assemblies (Figure 6e).”

“Of these, 808 and 845 are bubbles with haplotype path information for each assembly, and so are amenable to association investigations. Several of these genes are listed in the OMIA (Online Mendelian Inheritance in Animals) database as they harbour alleles causing phenotypic variation in cattle”. Is this more or less than you would expect by chance? Can you predict the coding consequence of the changes e.g. with VEP? Are the SVs in genes intolerant to change e.g. with high or low pLI scores of their human orthologues? I would hypothesize these genes are disproportionately those where a change is tolerated e.g. olfactory receptors etc but the author don't seem to list which genes they are anywhere?

We have expanded the results given on the set of coding-overlapping SVs, as well as including Supplementary Table 8 which contains all overlapping genes and overlap coordinates and length for both hifiasm and Shasta pangenomes (line 444). Genes matching those in the OMIA database are highly likely to be significant; however, there are only 168 entries in OMIA for cattle with gene information, and so the 5 hits observed are in line with expectations (lines 449-452). We performed a gene ontology enrichment analysis, and found that immunomodulation or sensory genes were abundant. This is consistent with prior studies in cattle which identified an enrichment in copy number variants or other structural variation in these gene ontology categories (Bickhart et al., 2012). We further find a total of 41 genes that had pLI scores above 0.9, which may be prioritized in future studies (lines 446-449).

Previous:

“Of these, 808 and 845 are bubbles with haplotype path information for each assembly, and so are amenable to association investigations. Several of these genes are listed in the OMIA (Online Mendelian Inheritance in Animals) database as they harbour alleles causing phenotypic variation in cattle”

New:

“We identified 808 and 845 genes in the ARS-UCD1.2 genome annotation (Refseq release 106) whose coding regions overlap bubbles with complete haplotype information in hifiasm and Shasta based pangenomes respectively, with 722 common to both (Supplementary Figure 6, Supplementary Table 8). This overlap is less than expected under random distribution of the same bubbles (one-sided Wilcoxon test: $p=6.9e-10$, methods), as expected given the greater evolutionary conservation of genic regions. Out of 229 affected genes which had pLI scores in human orthologues (Fuller et al., 2019), 35 and 36 overlapping bubbles in hifiasm and Shasta pangenomes (30 common to both) exceeded the threshold of 0.9 indicating they are intolerant to change (Supplementary Table 8). Separately, five and three genes that overlapped with bubbles in hifiasm and Shasta pangenomes are listed in the OMIA (Online Mendelian Inheritance in Animals) database of 168 genes that cause phenotypic variation in cattle (Supplementary Table 8);”

Line 648: “Contigs were scaffolded into chromosomes by the reference-guided approach of RagTag (Alonge et al., 2019)”. That is the reference for RaGOO which RagTag supersedes. I believe the correct reference for RagTag is Alonge, Michael, et al. "Automated assembly scaffolding elevates a new tomato system for high-throughput genome editing." bioRxiv (2021).

<https://doi.org/10.1101/2021.11.18.469135>. Also needs correcting in results section.

We thank the reviewer for noting the release of the RagTag preprint, and we have updated the citations.

“Combined, these results support that cataloguing most within-breed structural variation is a tractable goal through de novo long read assembly of 50 samples per breed, sufficient to represent the effective population size.” I don’t see how came up with this number? Surely depends on various factors e.g. the breed being studied, the diversity within the breed, the diversity between the animals selected to be sequenced, the minimum allele frequency of the SVs you want to capture. I think unless can better justify this statement should be removed.

We have updated the final paragraph containing the discussion on effective population size to substantiate the claim with references and a more explicit argument (lines 567-576). We agree that the specifics of a study, like the minimum allele frequency of the SVs of interest, will influence the number, but we believe 50 to be a general guide for capturing most diversity in a typical cattle breed/population.

“Our work shows that advances in sequencing and algorithms enable computationally feasible haplotype-resolved assembly of 20x HiFi coverage while retaining 90+% accuracy of detecting SVs when integrated into pangenomes. Assuming an effective population size of 100 for most breeds of cattle (Stachowicz et al., 2011), we expect most structural variation of a population/breed can be captured in a pangenome with haplotype-resolved assemblies from 50 individuals that have been prioritized to maximize haplotype diversity (Ros-Freixedes et al., 2017). Extensive existing short-read sequencing data could then be leveraged to genotype SVs present in the pangenome (Ebler et al., 2020; Hickey et al., 2020) and then imputed into tens of thousands of cattle previously genotyped with microarrays, enabling structural variant analysis at hitherto unattainable scale.”

Minigraph is only one way of constructing pangenomes and it may be worth adding a sentence or two to the discussion regarding the other possible methods that could have been used and their advantages and disadvantages. For example although minigraph has advantages such as it is fast it does have limitations versus reference-free approaches such as cactus.

We’ve expanded the discussion mentioning alternative pangenome approaches to reinforce that minigraph is not the only pangenome builder and that alternatives like pggg and cactus offer greater resolution (and different analyses) at the expense of runtime (lines 551-554).

Previous:

“Alternatively, reference-free approaches to pangenomes (Armstrong et al., 2020) may also circumvent these issues.”

New:

“Alternatively, reference-free approaches that use base-level alignment may circumvent these issues and resolve variation down to single nucleotides (Armstrong et al., 2020; Garrison, n.d.), which may prove crucial for breakpoint resolution and related analyses.”

Availability of data – are the authors not going to make the assembled genomes available somewhere?

All data used in this study to create the assemblies is publicly available, as well as the pipelines used, allowing our results to be reproduced. We've made the assemblies from Table 1 and Supplementary Table 3 available through Zenodo (line 158).

Reviewer #2 (Remarks to the Author):

Review – Leonard et al., Bovine pangenome reveals trait-associated structural variation from diverse assemblies

This manuscript describes the thorough investigation of 5 de novo assemblies using 2 different third generation sequencers and several different methods to create haplotype-resolved assemblies. The authors compared the performance of phasing between different levels of heterozygosity in the F1. They explored the minimal sequencing depth needed to reach given standards, by down-sampling the sequence data. Next, they created pan-genomes in different combinations and with different reference genomes as backbone, to compare phylogeny and to identify genes located in structural variation bubbles that may be trait associated.

It is a well written and structured manuscript, my compliments to the authors to present all the results of the many different methods applied in a concise way. However, I have 2 general comments and a number of line-by-line comments.

We thank the reviewer for their time spent on our manuscript and their feedback which helped us improve our manuscript. We have taken care to address both the 2 general comments as well as the line-by-line comments.

General comments

1) I am missing a conclusion at the end of the discussion.

We have rewritten the final paragraph to be a better concluding statement, indicating how 20x HiFi coverage is sufficient to produce haplotype-resolved assemblies which can accurately reveal SVs through a pangenome, and how this will enable a new scale of SV analysis across populations/breeds (lines 567-576). See response to reviewer #1 for further details.

2) For the trait associated SV results, it is not very clear which steps were taken to prioritize the presented results and it is not clear what portion of the detected SVs in the pan-genomes is (known to be) trait associated. The title puts a lot of emphasis on this part, and I missed a sort of overview from which the examples were picked to explain in more detail. The authors start off with 2 specific SVs, but it is not clear why these were chosen to report on in detail. They are certainly of interest, but please guide the reader. Thereafter, ~900 genes are mentioned of which several are in OMIA, but it is not clear how many. Also, how the set of genes in the last paragraph (starting at L453) were prioritized is not given. Were there cross references with other databased besides OMIA? Taken these comments together it would be nice to get a better feeling of how many SVs were detected, and how many are (known/likely to be) trait associated, in addition to the ones that are described in detail. Also, I suggest to add (to Material & methods) a section to explain how you declared your detected SVs as trait associated.

We have added an introductory sentence to the trait-associated SV section, giving a better overview of the motivation and how we selected the set of SVs we investigate in further detail (lines 387 and 463).

Previous:

“Pangenome integration of haplotype-resolved assemblies representing multiple breeds/species supported investigation of the evolutionary history of a multi-allelic copy number variation...”

New:

“We selected several previously identified SVs known to impact phenotype which were present in our pangenomes for further investigation”

“We also investigated genes in this set previously reported to harbor variants that affect phenotype in cattle.”

We address the other points on the requested details in a response to reviewer #1.

Line by line comments

L61-63 I suggest to reformulate this sentence, because larger SV, like CNV, can be and have been detected using short read sequencing in cattle (e.g. Mei et al. 2020; Butty et al., 2020; Kommadath et al., 2019). Of course repetitive regions and reference genome quality hamper SV detection using short reads and needs to be mentioned. Moreover, given the title, the SV topic is rather limited covered in the introduction. It may be highlighted that third generation technology can be beneficial in complex regions.

We rephrased the introduction to clarify previous studies have used short reads to examine SVs, including the references recommended by the reviewer. We additionally added text to discuss the value of long reads in analysing SVs and to highlight that SVs may be more important than previously estimated in eQTLs (lines 59-66).

Previous:

“Larger structural variants (SVs) and variation located in repetitive or challenging regions have rarely been studied across Bovinae due to the inherent limitations of short sequencing reads and incomplete reference genomes.”

New:

“Short sequencing reads have also been used to study larger structural variants (SVs) and variation located in repetitive or challenging regions across Bovinae (Butty et al., 2020; Kommadath et al., 2019; Mei et al., 2020), although their accuracy is limited compared to using long sequencing reads (Mahmoud et al., 2019). SVs may be involved in expression quantitative trait loci more often than previously estimated, and can impact gene expression more than shorter variants (Chiang et al., 2017). However, previous studies

suffer from potential reference bias resulting from the use of a single taurine cattle reference assembly.”

L113 add '(SR)' behind short reads to make clear this is what the abbreviation in the Figure 1 stands for.

We have included "(SR)" as suggested to make the abbreviation clearer (line 111).

L139 remove 'Table 1' at start of sentence

We removed the erroneous "Table 1".

L170-173 It is not clear to me how you can separate the 2 facts, lower coverage and lower heterozygosity compared to the other F1's, in this matter. Could they not both have an effect on PG50?

We have rephrased this to indicate that the limited coverage affects our ability to trio bin the ONT data and assemble haplotype-resolved reads (would be approximately 16x coverage each, which produces assemblies with NG50 in Kbs) rather than a gradual drop in PG50 from decreasing coverage. Lower heterozygosity does play a role in the decrease in PG50, but in this instance the change of assembly approach is the predominant driver (lines 171-174).

Previous:

The lower PG50 of the Original Braunvieh Shasta assembly reflects limited ONT coverage that resulted in the need to perform diploid polishing, rather than directly assembling trio binned reads, and is not due to an inherent limitation imposed by the level of heterozygosity.

New:

“There was insufficient ONT coverage for the OxO to perform the bin-then-assemble approach used for the NxB and GxP and so instead a diploid polishing approach was used to phase the haplotypes (methods), resulting in reduced PG50.”

PG50 is fairly consistent with NG50, indicating that phasing depends on heterozygosity and, specifically for ONT data, sufficient coverage to trio bin the reads, rather than increasing depth of coverage. P/NG50 can be sensitive to small fluctuations in contig length distribution, hence some of the variance seen in the figure below, but overall shows that lower coverage affects PG50 similarly to NG50.

Figure r1. PG50 decreases with coverage for both hifiasm and Shasta in a comparable decrease to NG50 (left), as seen by the fairly consistent ratio of PG50/NG50 across different coverages (right).

L175-189 The generated HiFi assemblies are especially longer due to the centromeres and telomeres, I was wondering how much longer it is excluding centromere and telomere regions, as the remaining additional length is probably more interesting from a functional perspective, but may be limited.

For the O, N, B, G, and P respectively, there are 90, 110, 140, 40, and 80 Mb of additional autosomal sequence in hifiasm assemblies compared to Shasta assemblies. Of the additional sequence, 50, 77, 93, 14, and 45 Mb respectively come from centromeric sequence, covering $56 \pm 12\%$ of the additional sequence. We added the following text to line 181.

“especially centromeric repeat sequence which accounted for 56% of the additional sequence compared to Shasta assemblies.”

L201 not sure what is meant with routinely in this context

We have removed the unnecessary use of “routinely”.

L213 CLR not yet defined, please add

CLR (continuous long read) was previously defined in caption of Table 1 but we have now defined it in the main text (line 50) and removed the definition in the caption.

L223-234 It is a pity that PG50 is not included in Figure 3, that could back up your comment in L170-173. If a lower coverage in this more heterozygote cross would also show drop in PG50, it becomes more plausible that indeed the lower heterozygosity did not play a role in PG50 for OxO.

We refer to the previous response to PG50.

L383 Pausch et al. detected the QTL, however they didn't report the presence of a CNV, so either rephrase the sentence or remove the reference.

L386-387 also here the relevance of the reference is not clear to me, please adjust

We have removed the references as they were no longer relevant after restructuring the text (see response to reviewer #1).

L396-398 Did you have a look at the parental SR data, that may help in resolving the copy number

We were able to confirm that the predicted copy numbers are plausible given a population-wide distribution identified in other BSW/OBV cattle samples (Figure r2), but were unable to explicitly determine which predicted copy number (HiFi or ONT) was more likely as we were unable to unambiguously determine the number of copies for the parental haplotypes based on short reads. We have also clarified this in the revised text (lines 405-408).

Figure r2. Number of CNV copies estimated for a large cohort of BSW and OBV animals based on the short read coverage. We followed the same approach to determine the diploid copy number as reported in Fig 2C in (Lee et al., 2021).

L441 Please indicate how many are listed in OMIA, as several is rather vague

L453 How were these 'further putatively trait-associated regions' detected, did they also harbour genes in OMIA or from some other database?

We have extensively added details on how we identified SVs for further investigation and provided explicit quantification of the SVs via Supplementary Table 5 and 8 (lines 288 and 444). We have rephrased the introduction to "putatively trait-associated" to clarify that changes to these genes have previously been reported to have phenotypic effects, although

potentially in other breed/species or different coding regions (line 463). We refer to the response to reviewer #1 for additional details.

Previous:

“The bubbles in the pangenome indicate further putatively trait-associated regions.”

New:

“We also investigated genes in this set previously reported to effect phenotype.”

L457-459 Any reference to substantiate this finding (which doesn't seem to be part of this study)?

We have added a missing reference to the supplementary notes which expands the finding, as well as a new reference supporting the fact that loss of function alleles in bovine *QRICH2* are associated with drastic phenotypic consequences. The new changes are highlighted in red (lines 466 and 470).

“Loss of function alleles in mammalian *QRICH2* orthologs lead to multiple morphological abnormalities of the sperm flagella (Hiltbold et al., 2021; Shen et al., 2019). We find that the fifth exon of *QRICH2*, which is affected by the coding sequence expansion, is transcribed in high abundance (>30 transcripts per million) in testes of mature taurine bulls (Supplementary Notes).”

L532 T2T not defined

We have replaced “T2T” with “telomere-to-telomere” as there are no longer any other T2T references (line 550).

L545-546 what exactly is then the benefit if they are still difficult to resolve with de novo haplotype resolved assemblies and pangenomes?

We have clarified the sentence to show that assemblies and pangenomes are better at resolving these cases than long read alignments, even if the regions remain challenging. New text is highlighted in red (line 566).

“These cases highlight the benefits of *de novo*, haplotype-resolved assemblies and pangenome which are better able to resolve the variation.”

L553 this 50 seems a rather arbitrary number or is it based on effect population size of cattle breeds?

We have expanded this into our conclusion and substantiated the claim on why 50 individuals is an appropriate sample size (lines 569-573). We refer to the response to reviewer #1 for additional details.

L556-557 just a comment: LD between SV and SNP is low (as you indicate in L363) and hence SV may not be accurately imputed into array genotypes, but that needs to be investigated.

We thank the reviewer for the comment, this is indeed an open and important question to address in future work. While there is limited correlation between phylogeny predictions using SVs or SNPs, we believe there will be value in applying imputation techniques to genotyped-SVs.

L616 Please add what Peregrine was used for.

We have clarified that Peregrine is one of the genome assemblers we tested (lines 644-646).

“Peregrine assemblies (C. S. Chin & Khalak, 2019) (version main:2aefc14+) were produced using the settings “--with-consensus --shimmer-r 3 --best_n_ovlp 8” on either the trio binned or total HiFi data set where appropriate. Only the primary assembly was retained.”

L625 OBV? Think it was defined as O, please correct throughout manuscript.

We now refer to “Original Braunvieh” (and other breed/species) in full during the methods to remove any uncertainty (line 655).

L635-637 does this mean you have done both? Which one is reported/used in this study?

We have rephrased this to state we used PEPPER polishing for all ONT-based assemblers for consistency, and removed the incomplete comparison to the raven assemblies polished with the default two rounds of racon (lines 668-669 and 658-665).

Previous:

“Raven (Vaser & Šikić, 2021) (version 1.5.0) assemblies were obtained from nanopore reads with default parameters. Polishing with the default two rounds of racon (Vaser et al., 2017) resulted in slightly lower QV compared to *post hoc* polishing with PEPPER.”

New:

“Raven (Vaser & Šikić, 2021) (version 1.5.0) assemblies were obtained from nanopore reads with default parameters, except for “-p 0” to disable racon polishing (Vaser et al., 2017). Assemblies were instead polished with PEPPER for consistency with other ONT assemblers.”

L656 Maybe add here that with quality you refer to NG50 and with phasing to PG50 for clarification

We have simplified where we referred to “quality value” and “phasing” to “QV” and “PG50” so the terms are consistent with Table 1. Likewise changed “contiguity” into “NG50” for consistency (lines 673 and 695).

References

- Bickhart, D. M., Hou, Y., Schroeder, S. G., Alkan, C., Cardone, M. F., Matukumalli, L. K., Song, J., Schnabel, R. D., Ventura, M., Taylor, J. F., Garcia, J. F., Van Tassell, C. P., Sonstegard, T. S., Eichler, E. E., & Liu, G. E. (2012). Copy number variation of individual cattle genomes using next-generation sequencing. *Genome Research*, *22*(4), 778–790. <https://doi.org/10.1101/GR.133967.111>
- Cheng, H., Jarvis, E. D., Fedrigo, O., Koepfli, K.-P., Urban, L., Gemmell, N. J., & Li, H. (2021). *Robust haplotype-resolved assembly of diploid individuals without parental data*. <https://arxiv.org/abs/2109.04785v1>
- Chin, C.-S., Wagner, J., Zeng, Q., Garrison, E., Garg, S., Functammasan, A., Rautiainen, M., Aganezov, S., Kirsche, M., Zarate, S., Schatz, M. C., Xiao, C., Rowell, W. J., Markello, C., Farek, J., Sedlazeck, F. J., Bansal, V., Yoo, B., Miller, N., ... Zook, J. M. (2020). A diploid assembly-based benchmark for variants in the major histocompatibility complex. *Nature Communications* *2020 11:1*, *11*(1), 1–9. <https://doi.org/10.1038/s41467-020-18564-9>
- Koren, S., Rhie, A., Walenz, B. P., Dilthey, A. T., Bickhart, D. M., Kingan, S. B., Hiendleder, S., Williams, J. L., Smith, T. P. L., & Phillippy, A. M. (2018). De novo assembly of haplotype-resolved genomes with trio binning. *Nature Biotechnology*, *36*(12), 1174–1182. <https://doi.org/10.1038/nbt.4277>
- Li, R., Gong, M., Zhang, X., Wang, F., Liu, Z., Zhang, L., Xu, M., Zhang, Y., Dai, X., Zhang, Z., Fang, W., Yang, Y., Zhang, H., Fu, W., Cao, C., Yang, P., Ghanatsaman, Z. A., Negari, N. J., Nanaei, H. A., ... Jiang, Y. (2022). The first sheep graph-based pan-genome reveals the spectrum of structural variations and their effects on tail phenotypes. *BioRxiv*, 2021.12.22.472709. <https://doi.org/10.1101/2021.12.22.472709>
- Luo, X., Kang, X., & Schönhuth, A. (2021). phasebook: haplotype-aware de novo assembly of diploid genomes from long reads. *Genome Biology*, *22*(1), 1–26. <https://doi.org/10.1186/S13059-021-02512-X/TABLES/3>
- Mamat-Hamidi, K., Hilmia, M., Idris, I., Di Berardino, D., & Iannuzzi, L. (2012). Chromosome evolution of the Malayan gaur (*Bos gaurus hubbacki*). *Firenze University Press*, *65*(1), 34–39. <https://doi.org/10.1080/00087114.2012.678085>
- Nurk, S., Koren, S., Rhie, A., Rautiainen, M., Bizkadze, A. V., Mikheenko, A., Vollger, M. R., Altemose, N., Uralsky, L., Gershman, A., Aganezov, S., Hoyt, S. J., Diekhans, M., Logsdon, G. A., Alonge, M., Antonarakis, S. E., Borchers, M., Bouffard, G. G., Brooks, S. Y., ... Phillippy, A. M. (2021). The complete sequence of a human genome. *BioRxiv*, 2021.05.26.445798. <https://doi.org/10.1101/2021.05.26.445798>
- Porubsky, D., Ebert, P., Audano, P. A., Vollger, M. R., Harvey, W. T., Marijon, P., Ebler, J., Munson, K. M., Sorensen, M., Sulovari, A., Haukness, M., Ghareghani, M., Lansdorp, P. M., Paten, B., Devine, S. E., Sanders, A. D., Lee, C., Chaisson, M. J. P., Korbel, J. O., ... Marschall, T. (2020). Fully phased human genome assembly without parental data

using single-cell strand sequencing and long reads. *Nature Biotechnology*, *39*(3), 302–308. <https://doi.org/10.1038/s41587-020-0719-5>

van Rengs, W. M., Schmidt, M. H.-W., Effgen, S., Wang, Y., Usadel, B., & Underwood, C. J. (2021). A gap-free tomato genome built from complementary PacBio and Nanopore long DNA sequences reveals extensive linkage drag during breeding. *BioRxiv*, *1*, 105–112. <https://doi.org/10.1101/2021.08.30.456472>

Wang, K., Hu, H., Tian, Y., Li, J., Scheben, A., Zhang, C., Li, Y., Wu, J., Yang, L., Fan, X., Sun, G., Li, D., Zhang, Y., Han, R., Jiang, R., Huang, H., Yan, F., Wang, Y., Li, Z., ... Kang, X. (2021). The Chicken Pan-Genome Reveals Gene Content Variation and a Promoter Region Deletion in IGF2BP1 Affecting Body Size. *Molecular Biology and Evolution*, *38*(11), 5066–5081. <https://doi.org/10.1093/MOLBEV/MSAB231>

Zanini, S. F., Bayer, P. E., Wells, R., Snowdon, R. J., Batley, J., Varshney, R. K., Nguyen, H. T., Edwards, D., & Golicz, A. A. (2021). Pangenomics in crop improvement—from coding structural variations to finding regulatory variants with pangenome graphs. *The Plant Genome*, e20177. <https://doi.org/10.1002/TPG2.20177>

REVIEWER COMMENTS

Reviewer #1 (Remarks to the Author):

Thank you to the authors for their response to the previous points raised. However, there are still some outstanding points.

Regarding the authors new text starting at line 521 “Currently there are no approaches to co-assembling HiFi and ONT data due to different read properties”. This isn’t actually true. For example see the WENGAN paper (<https://www.nature.com/articles/s41587-020-00747-w>) where they say “Furthermore, we show that the WENGAN assemblies performed by combining ultralong Nanopore reads with short or HiFi reads surpass the contiguity of the current human reference genome.” The WENGAN authors demonstrate their combined ONT and HiFi assembly (that they label HiFi+UL) produced a superior assembly to most approaches, including many also adopted by Leonard et al., and the point raised previously was it would be interesting to see if this extends beyond their paper, data and species, which this study seems well placed to address. Especially as have already run and compared many single technology assemblies.

I'm sorry but I think the new title may be more confusing than the previous one. Saying “twelve haplotype-resolved de novo assemblies of disparate origins” makes it sound like the twelve assemblies have disparate ancestral origins. This though isn’t the case, given 6 of them are effectively identical to the other six and are from the same animals and represent the same haplotypes. Unless by disparate origins the authors mean from different sequencing technologies? i.e. the two technologies used? Either way think has the potential to confuse. To address my previous concern that a set of five assemblies doesn’t provide the bovine pangenome I guess even just putting an “A” in front of the old title would at least indicate it is just one of potentially many “pangenomes” i.e. “A bovine pangenome reveals trait-associated structural variation from diverse assembly inputs”.

Given both reviewers flagged as a potential issue I was a bit surprised that the authors are maintaining the sentence “we expect most structural variation of a population/breed can be captured in a pangenome with haplotype-resolved assemblies from 50 individuals that have been prioritized to maximize haplotype diversity” I still think this would need to be explored properly rather than coming up with what on first appearance appears to be a pretty arbitrary number. A quick back of the napkin calculation would suggest that the majority of SVs would need to have a minor allele frequency greater than 0.7% for this to be true ($1-(1-0.007)^{100}=0.50$ i.e. an SV would have to have an allele frequency of 0.7% to have a 50% chance of being detected in 50 random individuals). But as far as I am aware we don’t know if this is the case? We do know though that the overwhelming number of genetic variants are rare (e.g. see panel A here, admittedly for humans <https://www.nature.com/articles/nature15393/figures/7>). Note the log scale on the y axis. SVs are even more likely to be shifted towards low frequencies due to their increased likelihood of being purged from the population by selection (see comment about pLi). At its most extreme, if every SV was removed from the population after 1 generation by selection then every SV would be a singleton, the number of

SVs would be total (not effective) population size*SV mutated genomes per generation, and the number of animals that would need to be sequenced to capture the majority of SVs would be $0.5 \times \text{total (not effective) population size}$. Although this is an extreme scenario my point is I don't see how you can just half N_e to know how many animals to sequence to get the majority of SVs as the authors suggest, even if we set aside the large variation in N_e across global breeds. If the authors really do want to keep this statement (which seems unnecessary) I think it needs to be made clearer why one can just half N_e in this way without accounting for the strong selection pressure on SVs and SV mutation rates etc. I'm being a pain about this as people may read this number and base future study designs on it so I believe it really needs to be more clearly justified as correct.

Sorry, just a clarification, my previous comment regarding pLI scores was actually the difference between the number of genes with high pLI scores versus the number expected by chance (randomly selecting the same number of genes) would be interesting as it would indicate the importance of SVs i.e. the missing difference will indicate the strength of selection against SVs (the missing ones being where selection has removed them). The bigger the difference the more important SVs arguably are. The authors have more focused on the ones left behind, which are arguably less important in that they have escaped being purged from the population, but I hadn't made this sufficiently clear and whether this is revised or not am happy to leave as just whether the authors think of interest.

Reviewer #2 (Remarks to the Author):

Although all my comments have been addressed, I still struggle with the balance/description of the topics in the paper.

The authors commented in the rebuttal: 'The reviewer states the two parts of the manuscript are (i) assessing genome assembly strategies and (ii) SV analysis. However, the two parts of the manuscript are better characterized as (i) pangenome graph construction from disparate assembly types/underlying data and quality comparisons of these graphs followed by (ii) comparison of these pangenomes for application to SV detection.'

I was under the same impression as reviewer 1 when I reviewed the paper, and therefore the SV part seemed underrepresented. Having read the rebuttal and reread the paper, I agree with the description given by the authors here of their own work, however that description is not made clear in the paper, starting with the aim and title. In the last paragraph of the introduction the SVs are not mentioned. And, the title really suggests you have been searching for SV with a trait association. While this description in the rebuttal suggest you were more after confirming the methods can find known complex SV.

So given the authors comment about the intention they have with this paper, there should be some parts rewritten to reflect that, to assure the paper matches the readers expectation.

REVIEWER COMMENTS

Reviewer #1 (Remarks to the Author):

Thank you to the authors for their response to the previous points raised. However, there are still some outstanding points.

Regarding the authors new text starting at line 521 “Currently there are no approaches to co-assembling HiFi and ONT data due to different read properties”. This isn’t actually true. For example see the WENGAN paper (<https://www.nature.com/articles/s41587-020-00747-w>) where they say “Furthermore, we show that the WENGAN assemblies performed by combining ultralong Nanopore reads with short or HiFi reads surpass the contiguity of the current human reference genome.” The WENGAN authors demonstrate their combined ONT and HiFi assembly (that they label HiFi+UL) produced a superior assembly to most approaches, including many also adopted by Leonard et al., and the point raised previously was it would be interesting to see if this extends beyond their paper, data and species, which this study seems well placed to address. Especially as have already run and compared many single technology assemblies.

We thank the reviewer for their suggestion to investigate the performance of another assembler. It is true that WENGAN accepts both ONT and HiFi data input, but the corresponding “ccsont” switch preset calls the routine Minia3, which then breaks those reads down into synthetic short reads and performs short read-based de Bruijn graph assembly. Therefore, it is not accurate to say that WENGAN works by “co-assembling HiFi and ONT data”, as we carefully stated in the manuscript, given that a short read-based approach used could never leverage the full power of highly accurate + long HiFi reads or ultralong ONT reads. Furthermore, this likely explains why WENGAN has poor reconstruction of highly repetitive regions (Supplementary Table 6 of WENGAN paper), while we demonstrate that assemblers like hifiasm, which natively exploits HiFi reads, excel at centromeric assemble. We also note that the WENGAN paper explicitly states that specialised HiFi assemblers can produce superior assemblies, for example:

“The WENGAN (HiFi + UL) assembly reaches an NG50 of 80.64 Mb, which outperforms all aforementioned assemblers, except for the recently developed HiCANU”

As well as in in Figure 2a of the WENGAN paper. We further demonstrate that hifiasm outperforms HiCanu in terms of NG50 as well as having similar strengths in repeat assembly, quality value, etc, and so logically hifiasm will outperform WENGAN HiFi+UL despite being HiFi-only (the reported NG50 in the hifiasm paper is in fact 10% larger than WENGAN for the same CHM13 sample). We also note that WENGAN uses version 0.1 of Shasta, the earliest release of the tool. We use version 0.7, which has several listed improvements to contiguity (introduced in versions 0.5 and 0.6), and so the difference in NG50 (Shasta: 58.1 Mb versus WENGAN: 80.6 Mb) would also be improved. We also note that WENGAN has not had any non-documentation updates since May 2020, while other tools have continued to improve substantially with updating sequencing performance of both HiFi and ONT. Additionally, the WENGAN paper states that the NG50 of HiFi+UL outperforms the human reference genome (NG50 57.9 Mb). However, the T2T assembly (without the Y chromosome in v1.1), based on the CHM13 sample as WENGAN but different to GRCh38, has an NG50 of 154.3 Mb (a 90% increase compared to WENGAN). So fully exploiting the benefits of HiFi and ONT data can produce a superior assembly to the method used by WENGAN. The T2T assembly was built with semi-manual efforts but is under development into a similar pipeline (<https://github.com/marbl/verkko>). To our

knowledge, this will be the first true HiFi+ONT assembler, although this is still algorithmically closer to HiFi-only backbone assembly + complex region resolution with ONT data as the previously mentioned challenges of combing HiFi and ONT still apply.

We note that WENGAN has no capabilities to handle phased information for haplotype-resolved assemblies, which we demonstrate has significant advantages (can produce an assembly with effectively 10x haplotype coverage for hifiasm). Although trio binning is an effective strategy and WENGAN could produce haplotype-resolved assemblies through that approach, we have demonstrated it is a less efficient use of the same sequencing coverage. After testing WENGAN on our HiFi and ONT NxB F1 trio binned data (approximately 28x and 55x respectively), we found WENGAN required over 1600 CPU hours, 240 GB peak RAM, and 2.7 Tb of intermediate storage, which is 2.7x slower than hifiasm. In addition, all quality metrics (Table 1_rr) were inferior to hifiasm, with WENGAN underassembling significantly (as previously reported for HiFi+UL in Supplementary Table 4 of the WENGAN paper) and only reconstructing about 1.5 Mb of centromeres in total across the autosomes (compared to about 90 Mb for hifiasm).

A central consideration of our work was identifying assemblers which maintain exceptional quality while allowing high-throughput of assemblies for future large-scale projects, and so WENGAN is not suitable based on the above-mentioned reasons and we have not added WENGAN assemblies to our manuscript. We provide access to all our data (short, HiFi, and ONT reads) with this manuscript, and so individuals can choose whatever assembler they believe to be appropriate for their intended analyses.

Table 1_rr. Assembly quality metrics for NxB (HiFi + ONT) using WENGAN, providing metrics like manuscript Table 1.

Breed	Size (autosomal size)	Contigs (autosomal contigs)	NG50	PG50	QV	BUSCO (single copy)	Repeat
Nellore	2.56 (2.51)	1515 (647)	37.3	37.2	45.1	93.1 (91.6)	40.68
BSW	2.68 (2.53)	1351 (519)	55.2	55.1	44.7	95.8 (94.2)	41.27

For clarity and technical correctness, we have updated the text in question (line 521) to

“Currently there are no approaches to co-assembling HiFi and ONT data that fully exploit the different advantages of the reads; ...”

I'm sorry but I think the new title may be more confusing than the previous one. Saying “twelve haplotype-resolved de novo assemblies of disparate origins” makes it sound like the twelve assemblies have disparate ancestral origins. This though isn't the case, given 6 of them are effectively identical to the other six and are from the same animals and represent the same haplotypes. Unless by disparate origins the authors mean from different sequencing technologies? i.e. the two technologies used? Either way think has the potential to confuse. To address my previous concern that a set of five assemblies doesn't provide the bovine pangenome I guess even just putting an “A” in front of the old title would at least indicate it is just one of potentially many “pangenomes” i.e. “A bovine pangenome reveals trait-associated structural variation from diverse assembly inputs”.

We thank the reviewer for their comments. In order to address any confusion of using “disparate” over the origin of animals or sequencing technology with respect to the twelve assemblies, and in line with comments from reviewer #2, we have suggested a title change to

“Structural variant-based pangenome construction has low sensitivity to variability of haplotype-resolved bovine assemblies”

We believe this addresses all concerns while accurately reflecting the intentions of the paper.

Given both reviewers flagged as a potential issue I was a bit surprised that the authors are maintaining the sentence “we expect most structural variation of a population/breed can be captured in a pangenome with haplotype-resolved assemblies from 50 individuals that have been prioritized to maximize haplotype diversity” I still think this would need to be explored properly rather than coming up with what on first appearance appears to be a pretty arbitrary number. A quick back of the napkin calculation would suggest that the majority of SVs would need to have a minor allele frequency greater than 0.7% for this to be true ($1-(1-0.007)^{100}=0.50$ i.e. an SV would have to have an allele frequency of 0.7% to have a 50% chance of being detected in 50 random individuals). But as far as I am aware we don’t know if this is the case? We do know though that the overwhelming number of genetic variants are rare (e.g. see panel A here, admittedly for humans (<https://www.nature.com/articles/nature15393/figures/7>)). Note the log scale on the y axis. SVs are even more likely to be shifted towards low frequencies due to their increased likelihood of being purged from the population by selection (see comment about pLi). At its most extreme, if every SV was removed from the population after 1 generation by selection then every SV would be a singleton, the number of SVs would be total (not effective) population size*SV mutated genomes per generation, and the number of animals that would need to be sequenced to capture the majority of SVs would be $0.5*$ total (not effective) population size. Although this is an extreme scenario my point is I don’t see how you can just half N_e to know how many animals to sequence to get the majority of SVs as the authors suggest, even if we set aside the large variation in N_e across global breeds. If the authors really do want to keep this statement (which seems unnecessary) I think it needs to be made clearer why one can just half N_e in this way without accounting for the strong selection pressure on SVs and SV mutation rates etc. Im being a pain about this as people may read this number and base future study designs on it so I believe it really needs to be more clearly justified as correct.

We agree with the reviewer that our statement in the concluding paragraph regarding the sequencing of 50 samples for SV detection is possibly too general and could potentially be misleading as such. This assumption was based on producing assemblies from carefully selected individuals that were prioritised to maximise haplotype diversity. These animals are “key ancestors” or “focal individuals”, and as such not random representatives for a breed. Key ancestor or focal individuals had previously been sequenced to assess an as large as possible sequence diversity from a small number of samples of founder populations (e.g., <https://pubmed.ncbi.nlm.nih.gov/22135348/>, <https://pubmed.ncbi.nlm.nih.gov/25017103/>, <https://pubmed.ncbi.nlm.nih.gov/23826801/>). These sample selection approaches typically prioritize animals that carry the most prevalent haplotypes and are thus less sensitive to detecting rarer variants (<https://pubmed.ncbi.nlm.nih.gov/29070022/>) or recent de-novo mutations. We still support our initial statement that 50 carefully selected samples are likely sufficient to capture most variants that segregate at a reasonable allele frequency (as the reviewer pointed out around 1%,

below which imputation accuracy will suffer (<https://pubmed.ncbi.nlm.nih.gov/28222685/>) in most breeds of cattle. Regardless, we have largely rewritten the paragraph to align its content better with our results and make clear that such an approach won't be able to discover the full spectrum of rare variants, particularly in breeds with larger effective population size. We also added references to emphasize that N_e is larger in indigenous/admixed than European breeds of cattle (<https://pubmed.ncbi.nlm.nih.gov/28219390/>, <https://pubmed.ncbi.nlm.nih.gov/32989325/>), and to direct the reader to a comprehensive study of SV diversity in a founder population (<https://pubmed.ncbi.nlm.nih.gov/33972781/>).

The last paragraph now reads (starting on line 569):

Our work shows that advances in sequencing and algorithms enable computationally feasible haplotype-resolved assembly of 20x HiFi or 60x ONT coverage while retaining 90+% accuracy of detecting SVs when integrated into pangenomes. Structural variation-based pangenomes built from these assemblies demonstrated significant consensus regardless of sequence platform and heterozygosity of the F1. Given the manageable read input needed, it is feasible to produce in the order of several dozens of haplotype-resolved assemblies for specific breeds of cattle. This effort can be de-centralized as no manual curation is needed to produce assemblies of sufficient quality for SV detection, even from purebred individuals. Due to the low effective population size of most breeds of cattle (Stachowicz et al., 2011), the resulting pangenomes would capture the most prevalent SVs within breeds, particularly when assemblies were produced from individuals that account for a large portion of the haplotype diversity of the population (Ros-Freixedes et al., 2017). More haplotype-resolved assemblies are required to reveal rare SVs (Beyter et al., 2021) and characterize SV prevalence in breeds with large effective population size or a history of admixture (J. Kim et al., 2017; K. Kim et al., 2020). Extensive existing short-read sequencing data could then be leveraged to genotype SVs present in the pangenome (Ebler et al., 2020; Hickey et al., 2020) and then impute them into tens of thousands of cattle previously genotyped with microarrays, enabling structural variant analysis at hitherto unattainable scale.

Sorry, just a clarification, my previous comment regarding pLI scores was actually the difference between the number of genes with high pLI scores versus the number expected by chance (randomly selecting the same number of genes) would be interesting as it would indicate the importance of SVs i.e. the missing difference will indicate the strength of selection against SVs (the missing ones being where selection has removed them). The bigger the difference the more important SVs arguably are. The authors have more focused on the ones left behind, which are arguably less important in that they have escaped being purged from the population, but I hadn't made this sufficiently clear and whether this is revised or not am happy to leave as just whether the authors think of interest.

We thank the reviewer for the clarification. For completeness, approximately 17.7% (3231 out of 18226) pLI scores are above the 0.9 threshold in the human dataset. We find 14.2% (42 out of 295) are above the threshold in our combined pangenome-SV analysis. We randomly sampled 295 entries from the human data 10,000 times and find only 5.85% have 42 or fewer genes with scores above the threshold (Figure 1_rr). This effect would be increased using the hifiasm-only, Shasta-only, or present in both pangenomes (35, 36, and 30 genes respectively), to as low as 0.06%. As such, it appears that there are fewer predicted loss-of-function SVs compared to random expectations.

However, this is just an estimate on a small (human orthologue) dataset, and should not be over-interpreted as an analysis on SV selection pressures in Bovinae.

Figure 1_rr: Random samplings of human pLI scores. Out of 10,000 iterations sampling 295 genes, 94.15% (blue) predicted more genes with pLI scores above the threshold than we observed in a bovine pangenome. The orange bars indicate iterations containing 42 or fewer genes with scores above the threshold, approximately 1.56 standard deviations from the mean.

Reviewer #2 (Remarks to the Author):

Although all my comments have been addressed, I still struggle with the balance/description of the topics in the paper.

The authors commented in the rebuttal: 'The reviewer states the two parts of the manuscript are (i) assessing genome assembly strategies and (ii) SV analysis. However, the two parts of the manuscript are better characterized as (i) pangenome graph construction from disparate assembly types/underlying data and quality comparisons of these graphs followed by (ii) comparison of these pangenomes for application to SV detection.'

I was under the same impression as reviewer 1 when I reviewed the paper, and therefore the SV part seemed underrepresented. Having read the rebuttal and reread the paper, I agree with the description given by the authors here of their own work, however that description is not made clear in the paper, starting with the aim and title. In the last paragraph of the introduction the SVs are not mentioned. And, the title really suggests you have been searching for SV with a trait association. While this description in the rebuttal suggest you were more after confirming the methods can find known complex SV.

So given the authors comment about the intention they have with this paper, there should be some parts rewritten to reflect that, to assure the paper matches the readers expectation.

We thank the reviewer for their comments. As addressed in a comment to reviewer 1, we have changed the title to

“Structural variant-based pangenome construction has low sensitivity to variability of haplotype-resolved bovine assemblies”

to clarify our intentions of this work and better connect to our results. We have reworked the abstract to better balance the two aspects of our work and state our aims more concisely. We have further rewritten several parts of the manuscript to further elucidate our aims and conclusions for this work.

Abstract (line 25):

We generated haplotype-resolved assemblies from the offspring of three bovine trios representing increasing levels of heterozygosity that each demonstrate a substantial improvement in contiguity, **completeness**, and accuracy over the current *Bos taurus* reference genome. **Diploid** coverage as low as 20x for HiFi or 60x for ONT was sufficient to produce two haplotype-resolved assemblies meeting the standards set by the Vertebrate Genome Project. **Structural variant-based pangenomes created from the haplotype-resolved assemblies** demonstrated significant consensus regardless of sequence platform, assembler algorithm, or coverage.

Introduction (line 79):

The present study **combines de-novo assembled** genomes from three bovine trios of varying heterozygosity **into structural variant-based pangenomes**.

Introduction (line 84):

This set of assemblies **from disparate sequencing and origin** is then used to evaluate the effects on pangenome construction depending on assembly approach,

Introduction (line 86):

The utility of a bovine pangenome is then demonstrated by analyses **using SVs to assess evolutionary relationships between Bovinae, recovering putatively trait-associated SVs, and quantifying SV-coding sequence overlaps**.

Discussion (line 570):

Structural variation-based pangenomes built from these assemblies demonstrated significant consensus regardless of sequence platform and heterozygosity of the F1.

REVIEWERS' COMMENTS

Reviewer #1 (Remarks to the Author):

I don't really want to keep going backwards and forwards on this but the authors previously stated that "currently there are no approaches to co-assembling HiFi and ONT data due to different read properties". When I then pointed out WENGAN they say this doesn't count because it makes synthetic short reads so isn't technically co-assembling HiFi and ONT data. I'm not sure I agree with them, in that it is taking both HiFi and ONT data and making an assembly. But even if they are right that WENGAN doesn't count this is just one example. They perhaps oddly highlight Verkko that they themselves says does co-assemblies but seem to suggest because it is an early release this doesn't count either. There are also further tools such as GALA that can also leverage data coming from both technologies. I guess maybe the authors think because it first generates dataset specific assemblies that it then combines this also doesn't count. But then I don't think they can say it is not taking advantage of the benefits of both technologies. I think the authors are going to have to have a very specific set of criteria to exclude all these tools while still saying there are no approaches to co-assembling HiFi and ONT data so I would still suggest they remove this statement.

Reviewer #1 (Remarks to the Author):

I don't really want to keep going backwards and forwards on this but the authors previously stated that "currently there are no approaches to co-assembling HiFi and ONT data due to different read properties". When I then pointed out WENGAN they say this doesn't count because it makes synthetic short reads so isn't technically co-assembling HiFi and ONT data. Im not sure I agree with them, in that it is taking both HiFi and ONT data and making an assembly. But even if they are right that WENGAN doesn't count this is just one example. They perhaps oddly highlight Verkko that they themselves says does co-assemblies but seem to suggest because it is an early release this doesn't count either. There are also further tools such as GALA that can also leverage data coming from both technologies. I guess maybe the authors think because it first generates dataset specific assemblies that it then combines this also doesn't count. But then I don't think they can say it is not taking advantage of the benefits of both technologies. I think the authors are going to have to have a very specific set of criteria to exclude all these tools while still saying there are no approaches to co-assembling HiFi and ONT data so I would still suggest they remove this statement.

We have removed the statement.